# Selective inhibition in CA3: A mechanism for stable pattern completion through heterosynaptic plasticity

**Gyeongtae Kim**  ¤, **Pilwon Kim** *

Department of Mathematical Sciences, Ulsan National Institute of Science and Technology (UNIST), Ulsan, Republic of Korea

¤ Current address: Center for Synaptic Brain Dysfunctions, Institute for Basic Science (IBS), Daejeon, Republic of Korea
* pwkim@unist.ac.kr

**Data availability statement:** All relevant code for this study is available on GitHub at

## Abstract

Engrams corresponding to distinct memories compete for retrieval in the CA3 region of the hippocampus, yet the detailed mechanisms underlying their formation remain elusive. Recent findings indicate that hippocampal inhibitory neurons display feature-selective firing patterns and diverse forms of synaptic plasticity, suggesting a crucial role in engram formation. Conventional CA3 attractor network models typically employ global inhibition, where inhibitory neurons uniformly suppress the activity of excitatory neurons. However, such models fail to capture the dynamics arising from sparse distributed coding or reflect inhibitory neurons' roles in the competition between engrams during memory retrieval. We propose a mechanism for engram formation in CA3 using a spiking neural network model that emphasizes heterosynaptic plasticity at excitatory-to-inhibitory (E-to-I) synapses. In our model, inhibitory neurons associate with specific neural assemblies during encoding and selectively inhibit competing engrams during retrieval. Driven by a simplified feedforward dentate gyrus (DG), this mechanism generates sparse, distributed engrams in CA3. This representation allows us to examine the effects of selective inhibition on pattern completion across various conditions, including partially overlapping engrams. Simulations show that selective inhibition substantially enhances recall stability and accuracy compared to global inhibition alone. Furthermore, emergent activity patterns across DG, CA3, and CA1 of the model replicate experimental signatures of pattern separation and completion. These results suggest that assembly-specific inhibition mediated by heterosynaptic plasticity could provide a parsimonious mechanism for engram formation and competition in CA3, offering testable predictions for future experiments.

## Author summary

We explored how memories are stored and retrieved in the hippocampus, focusing on region CA3, which plays a critical role in memory processes. Using a spiking neural

https://github.com/kgt1220/Hippocampus_SNN. The dataset supporting these findings is available on Zenodo at https://doi.org/10.5281/zenodo.14016721.

**Funding:** P. Kim was supported by the BK21 Program (Next Generation Education Program for Mathematical Sciences, 4299990414089) funded by the Ministry of Education (MOE, Korea) and National Research Foundation of Korea (NRF-2022R1A2C1092831) funded by the Korea government (MSIT). The funders had no role in study design, data collection and analysis, decision to publish, or preparation of the manuscript.

**Competing interests:** The authors have declared that no competing interests exist.

network model, we propose a novel mechanism whereby specific inhibitory neurons selectively control neural activity during memory retrieval. We found that this selective inhibition can naturally emerge during memory encoding. It provides an alternative to traditional models that rely solely on global inhibition and improves the stability and accuracy of memory recall. Moreover, our model aligns well with known cognitive functions of the hippocampus, helping explain complex memory processes, such as distinguishing between similar memories and accurately reconstructing past experiences. This research provides new insights into how detailed inhibitory neuron activity patterns formed during encoding shape memory retrieval, enhancing our understanding of memory formation and processing.

## Introduction

The hippocampus is crucial for processing episodic memories [1,2]. Within this region, memories are encoded and retrieved through engrams, physical traces instantiated by specific neural assemblies [3,4]. Sparse activity originating in the dentate gyrus (DG) gives rise to sparse, distributed CA3 assemblies [5]. This transformation supports pattern separation by converting highly similar inputs into distinct neural representations, thereby minimizing interference during retrieval [6–14]. Complementing this, the hippocampus can retrieve complete memories from partial or degraded cues via CA3. This reconstruction process, termed pattern completion [6,7,13–17], relies on two critical hippocampal pathways: the direct perforant path (PP) from the entorhinal cortex (EC) to CA3, and the recurrent collateral (Rc) connections within CA3. The direct PP conveys cues that enable engrams in CA3 to recognize similarities between incoming cues and stored memories. Rc connections are critical for CA3 to activate the full neural assembly, stabilizing the network and ensuring accurate memory retrieval. This characteristic underpins CA3's functionality as an attractor network [18–21].

Although pattern separation and completion processes in the hippocampus are well characterized, the underlying mechanisms that form engram in CA3, particularly the role of inhibitory interneurons, remain elusive. Existing attractor network models typically describe memory retrieval as a competition between CA3 engrams [18–21], yet they seldom address how those engrams emerge. Most previous models assume fully non-overlapping engrams and implement global inhibition, in which inhibitory neurons uniformly suppress excitatory activity irrespective of memory content.

Recent studies highlight the crucial roles of inhibitory neurons in memory processing [22,23]. Stimulus selectivity, the dependency of neuronal activity on external inputs, was primarily recognized in excitatory neurons [24,25]. However, emerging evidence indicates that GABAergic neurons demonstrate substantial stimulus selectivity [26,27]. Furthermore, inhibitory neurons exhibit diverse synaptic plasticity, including long-term potentiation (LTP) and long-term depression (LTD), similar to that observed in excitatory synapses [28–34]. This plasticity likely contributes to engram formation by modulating inhibitory control in response to experience. Extending beyond their known contribution to sparse distributed coding [35,36], these findings imply broader contributions of inhibitory neurons to memory encoding, storage, and retrieval.

Building on this evidence, we explore the role of CA3 inhibitory interneurons in memory processing using a spiking neural network model based on hippocampal architecture. We propose that CA3 inhibitory neurons can participate actively in memory retrieval through a mechanism we term selective inhibition. In this mechanism, inhibitory neurons

suppress excitatory neurons in competing engrams while sparing those within their associated engrams. We built our model to reflect the trisynaptic circuit from EC to CA3 via the DG, incorporating theta oscillation theory that distinguishes encoding and retrieval phases [37–39]. This allowed us to simulate how selective inhibition naturally emerges during engram formation under biologically inspired conditions. Specifically, the model includes heterosynaptic plasticity at E-to-I connections within CA3, drawing on evidence of heterosynaptic plasticity observed in various connections [40–43]. We demonstrate that heterosynaptic plasticity in this connection is crucial for the induction of selective inhibition during engram formation.

To investigate the influence of selective inhibition on memory retrieval, we represent the dentate gyrus as a feedforward layer. This simplification enables immediate, sparse distributed coding and creates varied competition conditions among engrams. The conditions explored include (1) unbiased competition between engrams (equal external drive), (2) different engram sizes, (3) multiple competing engrams, and (4) partially overlapping engrams. To examine each of these conditions, we first disabled heterosynaptic plasticity at E-to-I synapses (thus removing selective inhibition) and then compared retrieval performance with vs. without selective inhibition. We demonstrate that adding selective inhibition to global inhibition substantially improves retrieval stability across all conditions. We also show that our full network reproduces the expected pattern separation and completion activity across DG, CA3, and CA1, supporting the robustness of selective inhibition. Together, our results advance the understanding of how memory operations emerge from hippocampal circuitry, suggesting experimental measures to validate this mechanism.

## Results

### Model architecture and learning mechanisms

We built a spiking network model of the hippocampus that includes DG, CA3, CA1, and superficial/deep EC (Fig 1). The primary goal of this model is to evaluate the stability of pattern completion across a range of competition conditions. To do this, engrams in CA3 should be naturally formed for arbitrary inputs via the DG, resulting in a sparse distributed representation. However, modeling DG is challenging due to its intricate circuitry, complex synaptic connectivity, and heterogeneous neuronal types [44–46]. Therefore, we adopted a simplified approach by focusing solely on spatial pattern separation for non-sequential inputs, represented as 16-dimensional binary vectors at a fixed frequency ('Materials and methods'). Under these conditions, we designed a feedforward form of the DG, which we will explain in more detail in the next section, 'Feedforward form of the dentate gyrus and its inherent ability for spatial pattern separation.'

Critically, the model operates in two alternating phases tied to the theta cycle, implementing an idealized phase-dependent routing of information [37–39] (Fig 1). Each phase lasts 120 ms, reflecting a typical theta period and providing sufficient time for encoding or retrieval. Because there is no interaction between the encoding and retrieval phases in the model, we did not synchronize these phases with oscillatory cycles for most tasks, nor did we base oscillation generation on biological processes such as those in the medial septum [47–52]. This binary on/off gating deliberately simplifies experimental reality to emphasise phase-specific information flow; its implications are discussed later.

During the encoding phase, superficial EC projects only to DG and deep EC, which forwards signals to CA1, while DG drives CA3. In this phase, we silence CA3's direct PP input, Rc (except for inhibitory connections), Schaffer collateral (Sc), and the CA1 to deep EC connections (Fig 1A1). This prevents interference from previously learned memories, thereby

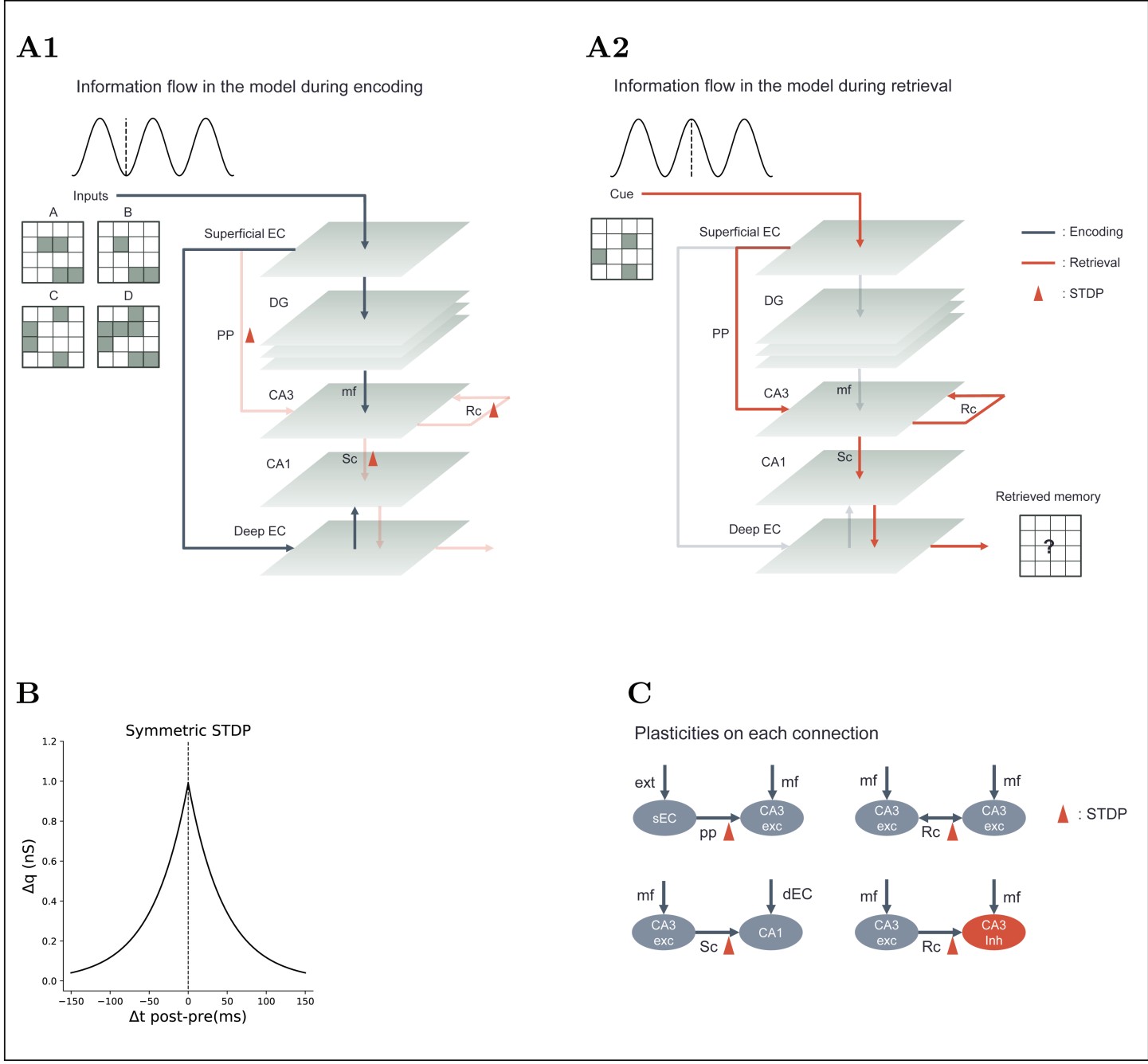

**Fig 1. Overview of model architecture and learning. (A)** The model includes major hippocampal subregions: DG, CA3, and CA1, along with the superficial and deep layers of the EC. It operates in two distinct theta-related phases: encoding and retrieval. **(A1)** Information flow during encoding. The superficial EC provides inputs to the DG and deep EC, but not directly to CA3. Key connections involved in CA3 (the PP from EC, Rc within CA3, and the Sc to CA1) are silenced during encoding, so synaptic plasticity occurs at those sites without triggering immediate CA3 output. **(A2)** Information flow during retrieval. The superficial EC sends the cue directly into CA3, which then reactivates the stored memory trace through Rc, CA1, and deep EC. **(B)** Symmetric STDP kernel used during learning. **(C)** Summary of synaptic plasticity across connections in the model. Abbreviations: ext: external input; mf: mossy fiber; exc: excitatory neuron; inh: inhibitory neuron.

maintaining the integrity of stored information while allowing new patterns to be encoded efficiently. Conversely, in the retrieval phase, the superficial EC sends the cue directly to CA3, which then reactivates the stored assembly via its recurrent collaterals, and the activity

propagates onward through CA1 and deep EC (Fig 1A2). Noise was introduced to CA3 excitatory neurons at a frequency of 3.5 Hz, following a Poisson distribution, which is an essential component for the attractor network [53]. This noise simulates the variability and spontaneous activity in biological neural networks, helping stabilize the attractor dynamics and prevent the network from getting stuck in suboptimal states.

Learning in the model is governed by biologically inspired plasticity rules. The direct PP, Sc, excitatory to excitatory (E-to-E) and E-to-I connections within CA3 continue to undergo symmetric spike-timing-dependent plasticity (STDP) (Fig 1B and 1C) [54–57]. Since these connections are silenced during encoding, learning is performed via heterosynaptic plasticity as described in the next section (Fig 1C).

## Engram formation with selective inhibition through heterosynaptic plasticity

This section details the mechanism of engram formation in the model, focusing on the heterosynaptic plasticity at E-to-I synapses in CA3. Mossy fibers from the DG project to both pyramidal neurons and inhibitory neurons in the CA3 region, facilitating sparse distributed coding [35,58–62]. We specifically had an interest in mossy fiber inputs to inhibitory neurons in CA3 due to their role in feedforward inhibition that regulates pyramidal neuron activity [4,63–68]. Our hypothesis goes beyond the conventional view of feedforward inhibition as merely a sparsification mechanism [35,36]. We propose that it also induces selective inhibition through heterosynaptic plasticity.

Heterosynaptic plasticity, where synaptic changes at one set of synapses affect another, is well-documented [40–43]. This form of plasticity occurs when the strength of a synapse changes due to activity at a nearby synapse. For example, activating one synapse can strengthen or weaken neighboring synapses, even if those synapses were not directly activated. This mechanism enables more complex forms of learning than traditional Hebbian plasticity. Multiple studies have demonstrated that heterosynaptic plasticity occurs in the direct PP, Rc, and Sc pathways [40–43]. However, most of the existing evidence pertains to excitatory neurons. In light of this, we hypothesize that heterosynaptic plasticity also occurs at E-to-I synapses in CA3. Building on this evidence and our hypothesis, the proposed model incorporates heterosynaptic plasticity at various types of synaptic connections in CA3 (Fig 1C).

During the encoding phase, processed outputs from DG stimulate excitatory and inhibitory neurons in CA3. Inhibitory neurons then suppress most of the excitatory neurons. The subset of excitatory neurons that remain active then forms an assembly with the concurrently active inhibitory neurons through heterosynaptic plasticity (Fig 2A). These active excitatory neurons also strengthen connections with simultaneously active neurons in the superficial Ec via the direct PP and CA1 via the Sc, forming the engram (Fig 1A1).

The proposed learning mechanism ensures that inhibitory neurons never suppress the excitatory neurons within their own engram (Fig 2C). Instead, inhibitory neurons can inhibit excitatory neurons in other assemblies, selectively dampening competing memories. Importantly, this rule does not confine an inhibitory neuron to a single engram. Rather, they exhibit stimulus selectivity. We quantified each CA3 neuron's storage capacity by computing the information it retained after encoding 500 arbitrary inputs. Fig 2B shows that ~85 % of excitatory neurons encode only one or two items, whereas inhibitory neurons span a wider range, with some storing many more. Thus, inhibitory neurons are information-selective yet capable of participating in multiple memories, in line with recent reports of stimulus-tuned interneurons [23,26,27].

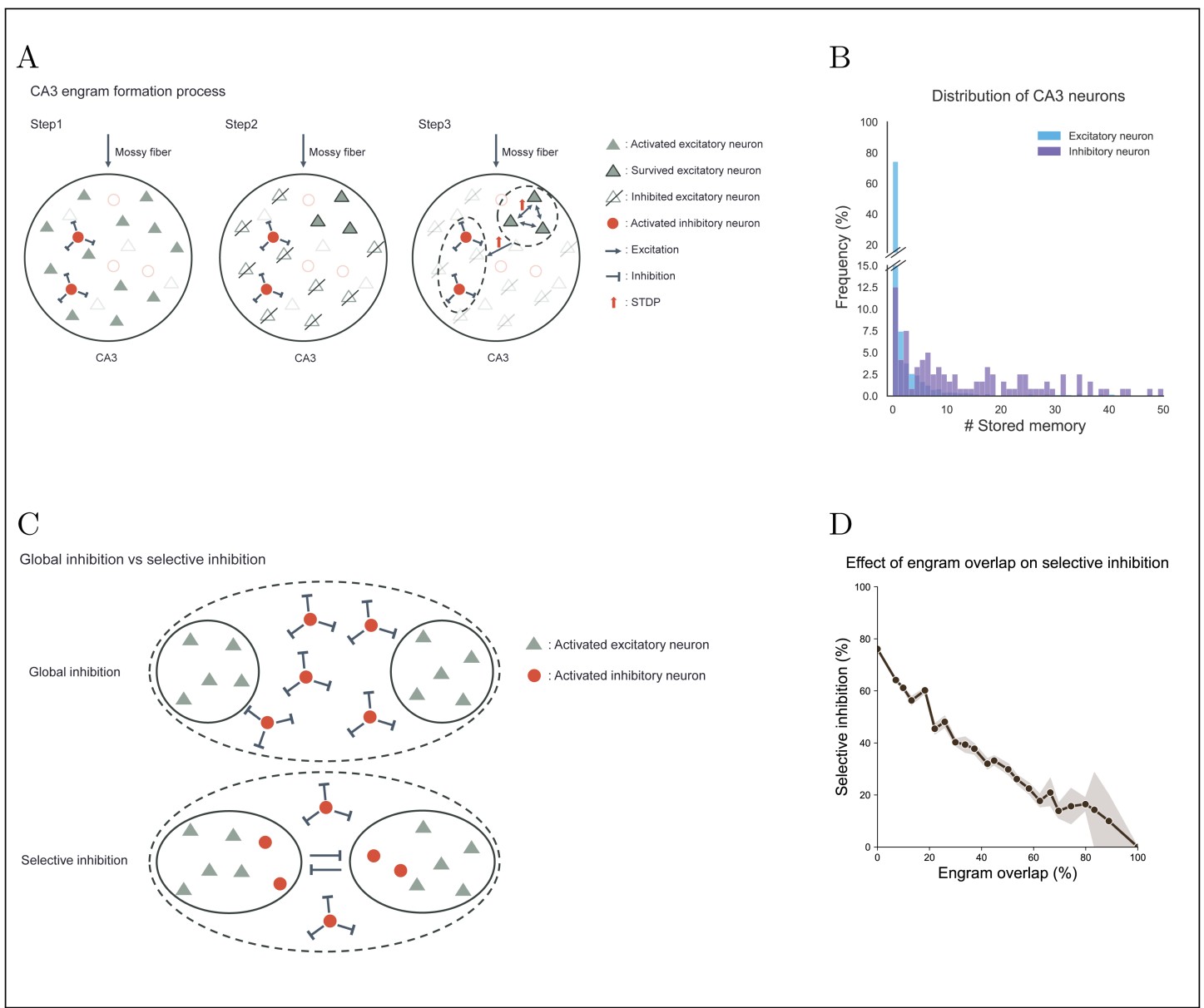

**Fig 2. Illustrations of CA3 engram formation inducing selective inhibition. (A)** The engram formation in CA3 driven by DG induces selective inhibition. Inhibitory neurons activated by mossy fibers inhibit most of the excitatory neurons. The surviving excitatory neurons and the activated inhibitory neurons form a neural assembly through STDP. **(B)** Distribution of CA3 neurons based on the number of stored memories. **(C)** Characterization of CA3 connections with inhibitory neurons for global versus selective inhibition. In the case of global inhibition, inhibitory neurons suppress excitatory neurons without regard to which engram they represent. In contrast, for selective inhibition, inhibitory neurons associated with a specific engram do not inhibit excitatory neurons within that same engram. **(D)** The inhibitory influence of one engram on others as a function of the degree of overlap between engrams.

To probe how engram overlap modulates selective inhibition, we measured the fraction of excitatory cells in one assembly suppressed by inhibitory cells from another across 500 simulated engrams. Engram pairs were binned into 25 overlap intervals (4% each). As Fig 2D illustrates, inhibitory influence declines steadily with rising overlap, showing that selective inhibition weakens when engrams share more neurons. This result underscores the need for sparse distributed codes in CA3 to preserve effective competition [6,7,13–15,17].

Strengthened E-to-I synapses make interneurons fire earlier during retrieval. These interneurons then silence substantial excitatory cells except those in their own assembly, steering the cue toward the correct memory. Because E-to-I weights start at non-zero values ('Materials and Methods'), classical global inhibition still emerges once recurrent excitation reaches threshold, supporting pattern completion.

We therefore view selective and global inhibition as complementary: targeted suppression minimizes interference, whereas recurrently driven global inhibition stabilizes network activity. To test this idea under realistic variability, we used a feedforward DG model to create CA3 engrams of different sizes and overlaps. The next section describes this DG construction and its performance in producing sparse, well-separated codes.

## Feedforward dentate gyrus and its role in instantaneous spatial pattern separation

To endow the DG so that it can perform pattern separation without learning, we modeled it as a feedforward network that preserves key anatomical characteristics while omitting intrinsic recurrent circuitry. Numerous modeling studies have demonstrated the effectiveness of DG in pattern separation [45,69–75], and a purely feedforward architecture is sufficient for this function [45]. In our model, the DG consists of two distinct layers: the hilus and the granule cell layer (GCL) (Fig 3). Although the biological DG contains several neuronal types, such as hilar interneurons, mossy cells, basket cells, and granule cells [46,76–78], we subsumed them into excitatory and inhibitory neurons (Fig 3).

Inhibitory neurons in the GCL receive two excitatory drives: direct projections from the superficial EC and input relayed through the hilus layer. The hilus pathway is further gated by local inhibitory neurons, providing the disynaptic inhibition critical for pattern separation [75]. We set the connection probabilities and synaptic weights manually while monitoring pattern separation performance ('Materials and Methods'). Although excitatory neurons in the hilus layer are inhibited, they also obtain a strong drive from superficial EC. Therefore,

Structure of the dentate gyrus in the model

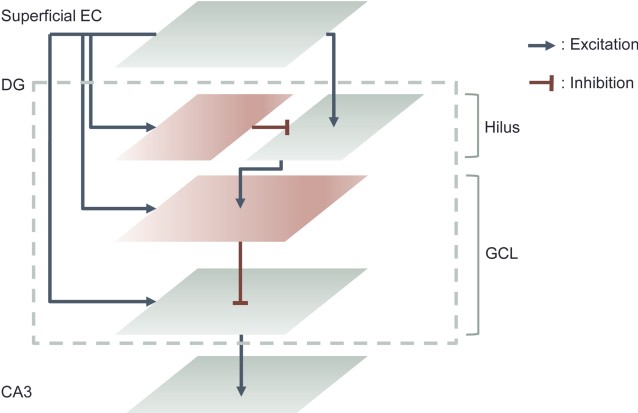

**Fig 3. Structure of the dentate gyrus in the model (A)** The DG is modeled as two feedforward layers: the hilus layer and the GCL. Each layer contains excitatory (gray) and inhibitory (red) neurons. Superficial EC inputs drive all DG neurons. Inhibitory neurons in the GCL also receive inputs from excitatory neurons in the hilus layer, which are regulated by hilus inhibitory neurons. Finally, excitatory neurons in the GCL, regulated by inhibitory neurons within the GCL, send mossy fiber outputs to CA3.

their net firing remains relatively high, keeping GCL inhibitory neurons active. These two excitatory inputs enable the GCL inhibitory population to detect key features of most input patterns (Figs 3 and 4A).

Excitatory neurons in the GCL fire only when they simultaneously escape this inhibition and receive excitation from the superficial EC. This mechanism transforms low-dimensional input into very sparse, high-dimensional representations. The resulting activation patterns of excitatory neurons in the GCL stimulate excitatory and inhibitory neurons in CA3 through mossy fibers, establishing sparsely distributed representations. The presentation of ten arbitrary input patterns to the dentate gyrus confirmed this behavior. Fig 4A shows that each pattern evokes a well-separated response in the excitatory GCL population.

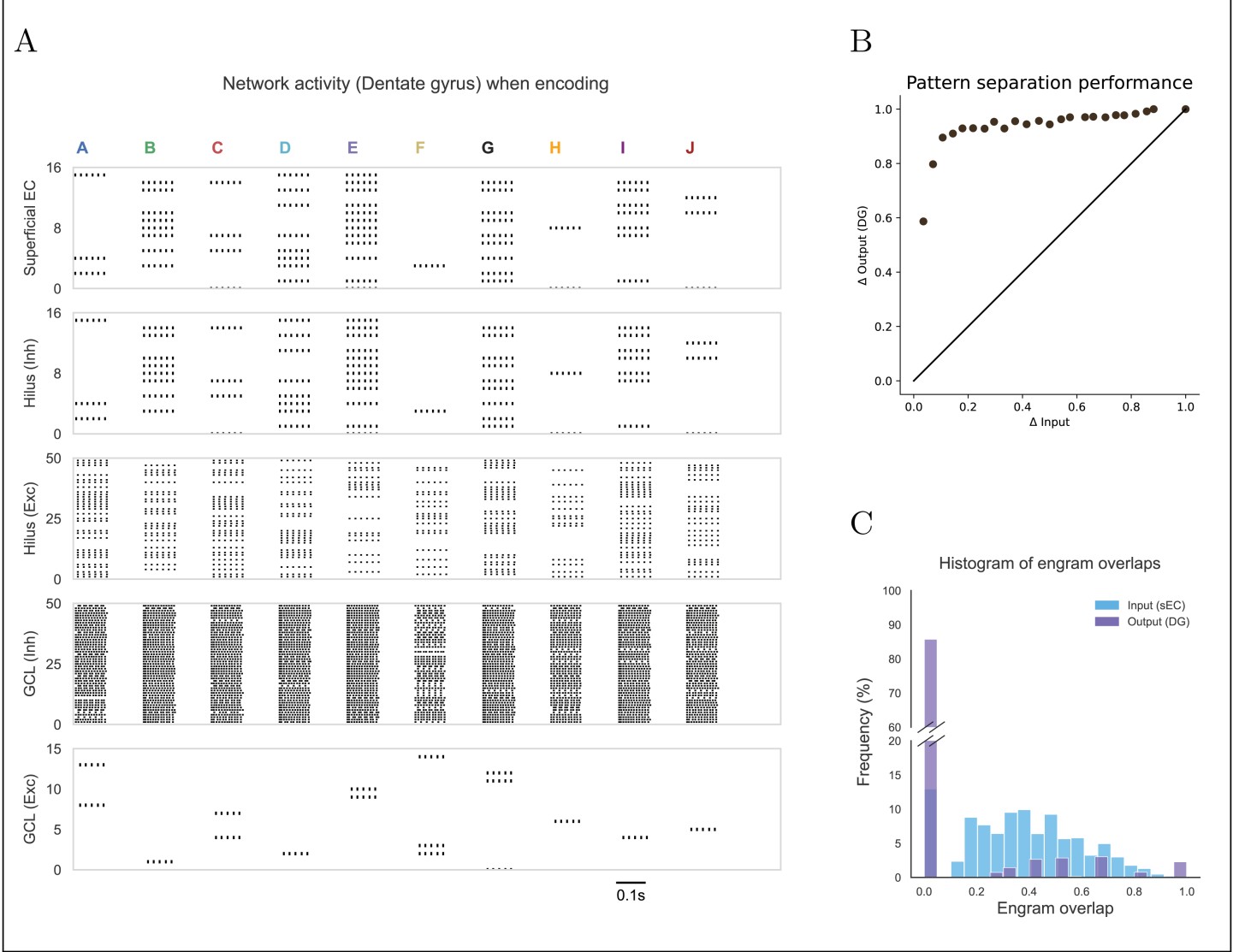

**Fig 4. DG activity and pattern separation.** (A) Spike raster during encoding of 10 distinct inputs (A to J). Plotted are neurons in the superficial EC, and excitatory and inhibitory neurons in both DG layers (hilus and GCL). 50 of the 100 hilus excitatory neurons and 50 of the 400 GCL inhibitory neurons are shown for visualization. The 16 excitatory GCL neurons encoding the 10 inputs (out of 800) are highlighted. (B) Pattern separation performance of the DG. (C) Histogram of engram overlaps: the similarity of DG output (GCL excitatory) engrams versus similarity of superficial EC input patterns. Most outputs fall into low-overlap bins, indicating robust separation.

To evaluate the pattern separation performance of our DG model, we presented 1,000 input patterns to the DG and recorded the activity of excitatory neurons in the GCL. For every pair of patterns, we computed their overlap with the metric of [79]; discrimination was defined as one minus the overlap value ('Materials and Methods'). Output discrimination indices were then averaged in 25 bins, each 0.04 wide (Fig 4B). When input discrimination exceeded ~0.1, the mean output discrimination rose sharply toward ~1.0 and remained high, indicating efficient separation. 4C plots two histograms: the similarity distribution of randomly sampled input pairs and the corresponding distribution for their output pairs. More than 85% of input pairs become fully separated, as shown by the large proportion of output pairs whose similarity falls into the lowest bin. Fig 4B and 4C show that the feedforward DG in the proposed model performs robust spatial pattern separation even in the absence of learning.

## Sparse distributed coding for memory storage and retrieval

With its feedforward dentate gyrus, the model separates cortical inputs and projects them into a high-dimensional space. The resulting activity supports sparsely distributed representations in CA3 and permits variance in engram size and in the degree of overlap among engrams (Fig 5A2 and 5D). To assess the storage capacity, we counted the number of distinct engrams that formed as we increased the number of input patterns (Fig 5C). The curve rises steadily by roughly 200 engrams as the input set grows from 600 patterns upward. It indicates that CA3 does not saturate within the tested range, confirming the efficiency of the DG-CA3 network in handling diverse inputs. Thus, the DG to CA3 cascade can accommodate larger input repertoires while preserving separation, making it suitable for testing the stability of pattern completion under the selective inhibition.

Additionally, we analyzed the correlation between the size of the input patterns and the size of their corresponding engram in CA3. Pattern size in each region was defined as the number of active excitatory neurons in the superficial EC and in CA3, respectively. Fig 5D reveals no correlation, indicating every CA3 engram contains about ten excitatory neurons, irrespective of how many EC neurons were active. This invariance implies that CA3 allocates its resources efficiently, forming stable assemblies that are insensitive to input characteristics.

Figs 5B and 6A illustrate the population activity evoked by 10 arbitrary inputs during the encoding and retrieval phases, respectively. Rather than changing individual synaptic weights, we updated the peak conductances ('Materials and Methods'). This approach keeps the synaptic weights for each connection fixed, thereby maintaining the connectivity ratio. For clarity, we refer to the learned weight as the product of the peak conductance and synaptic weight for each connection (Fig 5A).

During encoding, DG outputs elicit distinct firing patterns in GCL excitatory neurons, which subsequently activate excitatory and inhibitory neurons in CA3. With suppression from these inhibitory neurons, the DG output creates sparsely distributed patterns among CA3 excitatory neurons (Fig 5A2). Accordingly, excitatory-to-excitatory weights cluster along the diagonal of the weight matrix, reflecting strong recurrence within each assembly, whereas off-diagonal elements remain weak. In contrast, weight matrices that involve CA3 interneurons (Fig 5A3 and 5A4) show no obvious diagonal band, indicating that although these interneurons are input-selective (Fig 2B), they are not locked to any single engram.

To observe engram competition during retrieval, we repeatedly presented a cue that overlapped 50% with engrams E and G. Entering CA3 via the direct PP, the cue successfully reactivated either E or G in 80% of trials. Failures were defined as instances when no assembly achieved dominance, leading to interference and no deep EC output (Fig 6A). In successful

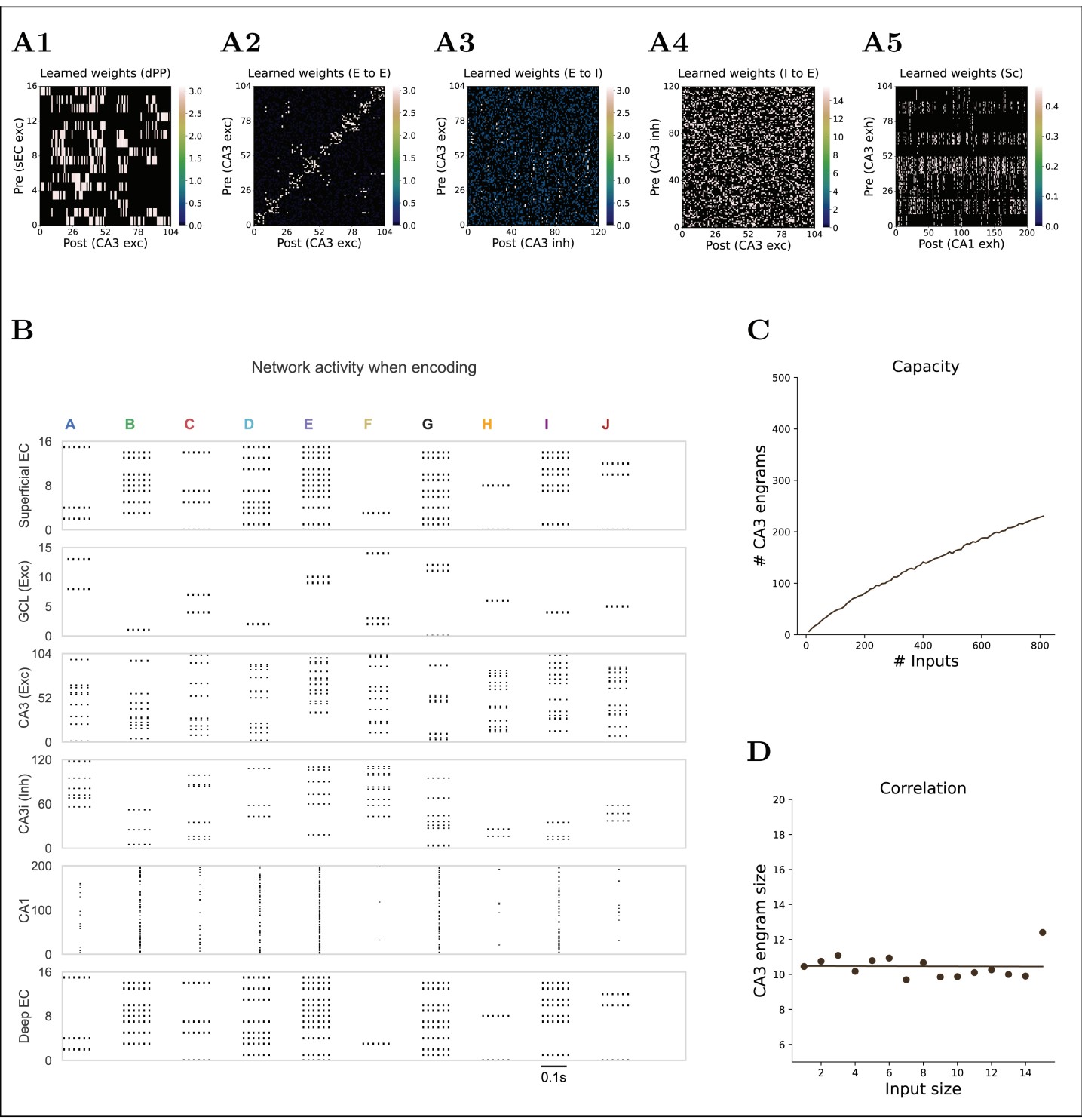

**Fig 5. Encoding of inputs and learned synapses.** (**A**) Weight matrices after STDP learning. Weights represent the connection weights multiplied by the learned peak conductances. Only CA3 neurons encoding the 10 example inputs are visualized; E, excitatory; I, inhibitory. (**A1**) Learned direct PP weight matrix from superficial EC to CA3 excitatory. (**A2**) Learned excitatory Rc weight matrix from CA3 excitatory to CA3 excitatory. (**A3**) Learned Rc weight matrix from CA3 excitatory to CA3 inhibitory. (**A4**) Learned Rc weight matrix from CA3 inhibitory to CA3 excitatory. (**A5**) Learned Sc weight matrix from CA3 excitatory to CA1. (**B**) Raster plot during the encoding of 10 different inputs (A to J). Shown are spikes in the superficial EC, excitatory neurons in the GCL, CA3 excitatory and inhibitory neurons, CA1, and deep EC. Other types of neurons in the DG are omitted. As in (**A**), only the CA3 excitatory neurons active in these examples are selected for visualization. (**C**) CA3 memory capacity: number of engrams formed versus number of input patterns. (**D**) Relationship between input pattern size and resulting CA3 engram size (points) with a linear fit.

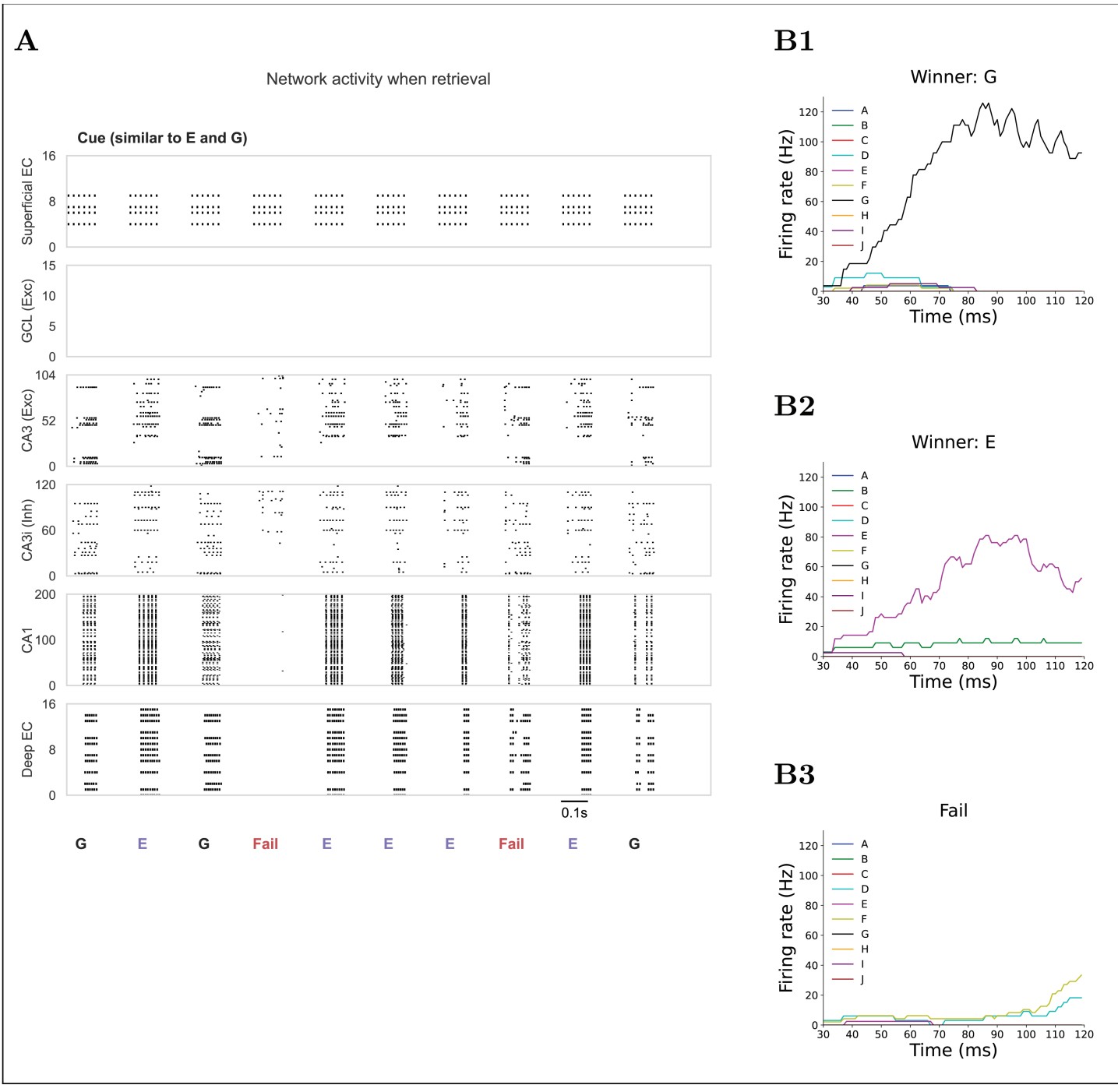

**Fig 6. Retrieval of learned memories. (A)** Spike raster during retrieval in response to a cue similar to inputs E and G out of the 10 encoded patterns A to J. Layer conventions are the same as in Fig 3B. The cue is presented in 10 trials; some yield successful recall of E or G (indicated by strong firing in deep EC for those patterns), while others fail. **(B)** CA3 engram firing rates during retrieval for the example trials. **(B1)** Successful retrieval of pattern G: the CA3 G-engrams dominate firing. **(B2)** Successful retrieval of pattern E: the CA3 E-engrams dominate. **(B3)** Retrieval failure: no single engram dominates, and activity remains low and diffuse.

trials, the winning assembly fired at ~80 to 120 Hz over a 30 ms window, whereas competing assemblies stayed below ~10 Hz (Fig 6B1 and 6B2). This suggests that the dominant assembly effectively suppressed the activity of competing assemblies. In contrast, no dominant activity

emerged in failure trials Fig 6B3). We note that such high firing rates are a model-driven outcome: in vivo CA3 pyramidal neurons are quiet (0.3 to 5 Hz) and emit only brief ~200 Hz bursts rather than sustained high-frequency trains [80]. In the model, the elevated rates enforce a clear winner-take-all attractor, ensuring that the active engram robustly suppresses its competitors.

Finally, we quantified the contribution of the direct PP. Varying percentages of partial cues (Fig 7A), silencing this pathway and routing the cue to CA3 solely through mossy fibers drastically reduced retrieval success (Fig 7C). Without the direct PP, the DG treats the degraded cue as novel, recruiting a distinct CA3 population via pattern separation. When the PP is intact, it directly excites assemblies whose stored features overlap with the cue, greatly improving recall across all levels of cue completeness (Fig 7B).

## Stability of pattern completion with selective inhibition under various conditions.

We demonstrated that CA3 in the proposed model encodes inputs in a sparsely distributed manner, along with sufficient engram capacity. This sparse coding scheme exhibits several properties influencing competition between engrams during retrieval. We identified four conditions affecting this competition: (1) cue strength disparity between engrams, (2) overlap

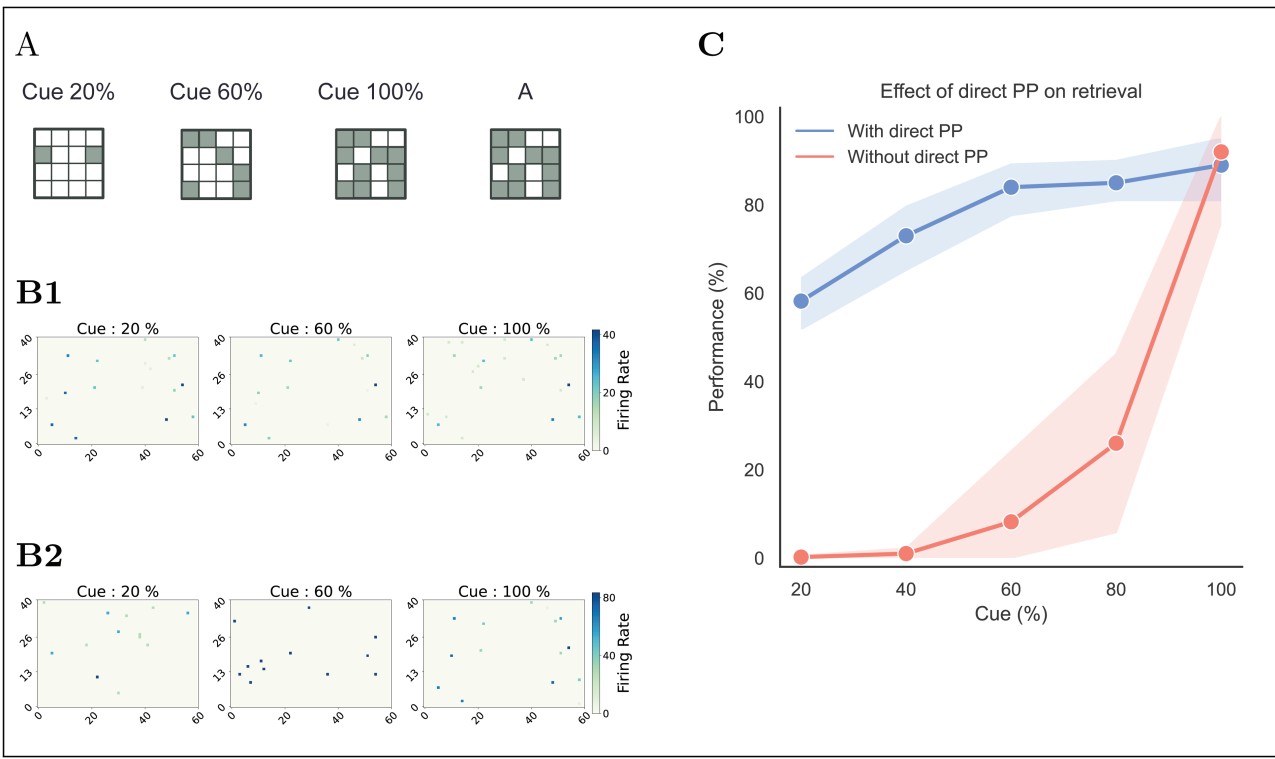

**Fig 7. Impact of the direct PP on retrieval. (A)** Illustration of different percentages of partial cues for a learned input A. **(B)** Firing rate heatmap for CA3 neurons in response to each cue during retrieval (120 ms). Each heatmap shows the most common firing pattern across 20 trials for that cue. **(B1)** With the direct PP intact, firing patterns are consistent across cue variations. **(B2)** With the direct path silenced, firing patterns vary strongly with each cue. **(C)** Retrieval performance versus percentage of cue under two conditions: with direct PP (blue) and without direct PP (red). Each point is the mean success rate (averaged over 5 random cues × 20 repeats; 100% cue uses one sample).

between engrams, (3) the number of engrams retrieved by a single cue, and (4) differences in engram size.

To investigate the effects of selective inhibition on retrieval under these conditions, we isolated this mechanism by disabling heterosynaptic plasticity at E-to-I synapses in CA3. This manipulation prevented the emergence of engram-specific inhibitory drive. Consequently, inhibition remained purely global, allowing us to compare retrieval performance with and without selective inhibition.

**Robust pattern completion with equal cue strengths.** According to the encoding specificity principle, retrieval is most effective when the cue closely matches the features present at encoding [81,82]. However, retrieval can become challenging when competing memories share significant similarities with the cue, leading to interference. Although such overlap can promote useful generalization, it is problematic when the memories must be kept distinct. A common way to limit interference is to bias the cue toward the target memory, thereby reducing its similarity to competing memories [83]. However, when two memories are highly similar, creating a meaningful bias becomes difficult and may lead to poor retrieval performance. This, in turn, diminishes the dentate gyrus's ability to discriminate highly similar inputs.

We therefore asked whether selective inhibition could stabilise pattern completion when a retrieval cue was equally similar to two competing engrams. To do this, we first identified two CA3 engrams, $M_1$ and $M_2$, and constructed an initial cue $C_0$ that shared exactly $k$ active superficial EC neurons with each ($|C_0 \cap M_1| = |C_0 \cap M_2| = k$; Fig 8A). We then introduced a bias toward $M_1$ by defining a bias level $b(0 \leq b \leq k)$ in which we replaced $b$ of the $k$ shared neurons in $C_0$ with $b$ neurons drawn exclusively from $M_1$, keeping the total cue size fixed at $k$. At bias level $b$, the cue contains $k-b$ neurons common to both memories and $b$ neurons unique to $M_1$. We applied this procedure to 15 distinct memory pairs whose CA3 engrams were equal in size and non-overlapping (Fig 8B and 8C). For each pair and each of the five bias levels, we performed 50 retrieval trials with and without selective inhibition.

Our simulations reveal a marked difference in retrieval accuracy with and without selective inhibition when cues match two CA3 engrams equally (Fig 8D). Without selective inhibition—i.e., under global inhibition alone—accuracy falls to about 60%, replicating the findings of Deco et al.'s model [83]. With selective inhibition, accuracy instead remains near 90% across all bias levels. This stable performance likely occurs because selective inhibition sharpens the contrast between competing memories, effectively amplifying small differences in input strength. These findings indicate that selective inhibition markedly enhances hippocampal pattern completion, preserving high retrieval success even with an unbiased cue.

**Pattern completion with overlapping engrams.** To test whether selective inhibition can safeguard recall when engrams overlap (Fig 8E, 8F, and 8G), we prepared 60 input pairs of equally sized CA3 engrams, whose overlap percentages varied from 0 to 0.3. For this simulation and the two that follow (size variation and competitor number), we used uniformly similar partial cues and ran 50 retrieval trials per condition. This design isolates the impact of overlap on pattern completion performance.

Retrieval accuracy declined as engram overlap increased, but performance with selective inhibition was consistently higher than performance without selective inhibition across all overlap percentages (Fig 8H). With selective inhibition, performance remained near 70% until overlap reached 0.1, whereas without selective inhibition, it had already fallen to approximately 30%. Both conditions exhibited sharp declines beyond 0.1 overlap. However, at the highest overlap tested (0.3), accuracy with selective inhibition remained around 35%, compared with substantially lower performance without selective inhibition. These findings suggest that although the benefit of selective inhibition diminishes as engrams share more

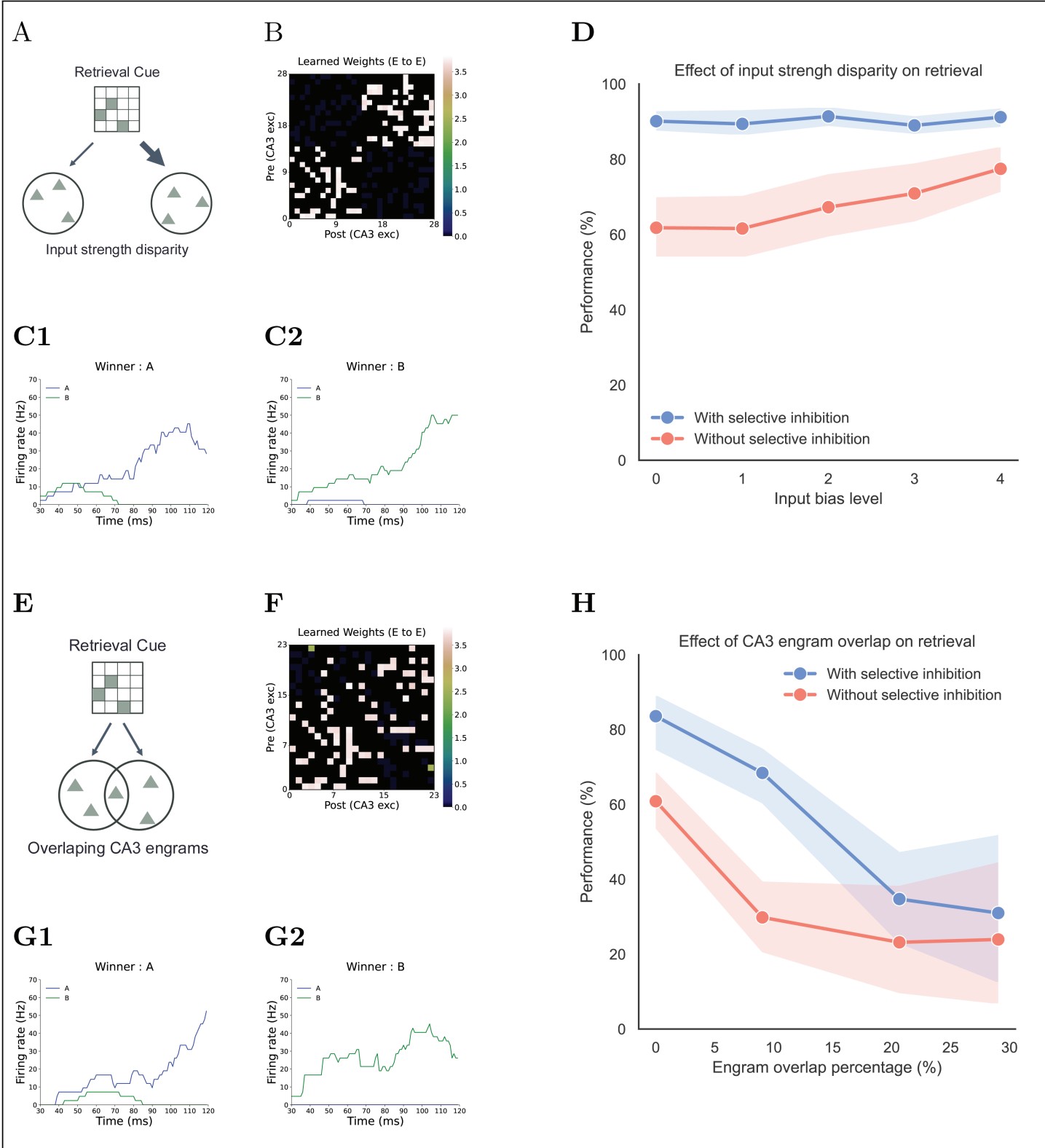

**Fig 8. Effects of input strength disparity and engram overlap on retrieval.** (A) to (D) Input strength disparity task: (A) Cue schematic: two CA3 engrams are cued with different input strengths. (B) Example of the learned Rc weight matrix. (C) CA3 engram firing rates during retrieval; two example trials with the same cue are shown: (C1) memory A is retrieved, (C2) memory B is retrieved. (D) Retrieval performance versus input strength disparity, comparing with (blue) and without (red) selective inhibition. Performance is measured as the mean retrieval success rate over 15 bias samples × 50 repeats each. (E) to (H) Overlap task: (E) Cue schematic: two

overlapping engrams are cued equally. **(F)** Example learned Rc weight matrix. **(G)** CA3 engram firing rates for example trials with the same cue: **(G1)** memory A is retrieved, **(G2)** memory B is retrieved. **(H)** Retrieval performance versus percentage overlap between engrams, comparing with (blue) and without (red) selective inhibition.

neurons, selective inhibition nonetheless reduces interference more effectively than relying on global inhibition alone.

**Pattern completion performance relative to the number of competing engrams.** We next examined how retrieval performance changes as more CA3 engrams compete for the same cue (Fig 9A). Although increasing the number of competitors naturally reduces

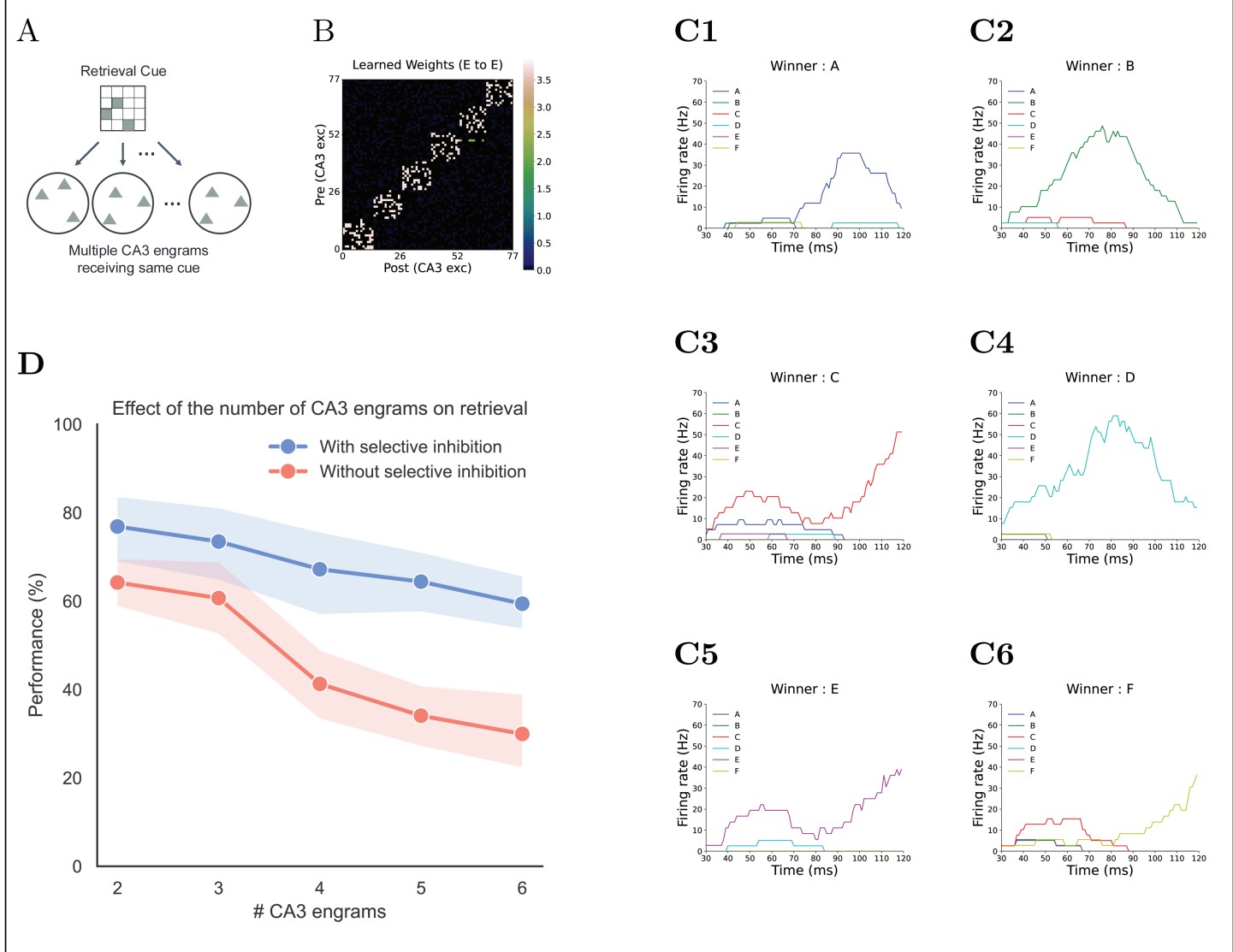

**Fig 9. Effect of the number of competing engrams on retrieval. (A)** SCue schematic: multiple CA3 engrams are presented with the same cue. **(B)** Example learned Rc weight matrix. **(C)** CA3 engram firing rates for example trials with the same cue: **(C1)** memory A is retrieved, **(C2)** memory B is retrieved, **(C3)** C **(C4)** D, **(C5)** E, **(C6)** F. **(D)** Retrieval performance versus number of engrams, comparing conditions with (blue) and without (red) selective inhibition.

performance [84,85], investigating this scenario allows us to assess how selective inhibition manages interference when multiple memories vie for retrieval. To explore this, we prepared 75 sets of non-overlapping, equal-sized CA3 engrams and varied the number of engrams per cue from two to six (Fig 9B and 9C). This design quantifies how selective inhibition counteracts the performance drop accompanying an increasing number of competing memories.

Fig 9D shows that retrieval performance declines as the number of competing engrams increases. With selective inhibition, retrieval success declined moderately from ∼80% with two engrams to ∼60% with six. By contrast, under only global inhibition, performance held around 60% with up to three competing engrams, then dropped steeply to ∼30% with six. These results indicate that selective inhibition mitigates the negative impact of increasing competition more effectively than global inhibition alone.

**Reduced sensitivity to engram size variations under selective inhibition.** In the model, the size of the engram in CA3 shows no correlation with input pattern (Fig 5D), suggesting that retrieval should depend on cue similarity rather than engram size; otherwise, larger engrams would dominate recall even if they match the cue less well. To test whether larger engrams gain an unfair advantage and whether selective inhibition can correct for size bias, we generated 75 input pairs stored in non-overlapping CA3 engrams with varying sizes (Fig 10A, 10B, and 10C).

The results show that with selective inhibition, performance stayed ∼80%, whereas without it, performance hovered around 60% (Fig 10D), mirroring the stable performance seen with equal cues (Fig 8D). To quantify bias in which memory was retrieved, we defined the retrieval ratio as the fraction of retrievals that returned the larger engram memory across the total number of successful retrievals. Under global inhibition alone, larger engrams accounted for roughly 0.3 of recalls, compared with about 0.2 when selective inhibition was present (Fig 10E). These results show that while engram size can skew retrieval under any inhibitory condition, selective inhibition attenuates size-based retrieval bias.

Our findings demonstrate that selective inhibition furnishes a robust mechanism for pattern completion under all tested engram competition scenarios. It improves retrieval despite differences in cue strength, reduces interference from overlapping engrams, maintains higher performance as more memories compete, and minimizes bias arising from differences in engram size. These results underscore the functional advantages of selective inhibition over global inhibition for reliable memory retrieval and therefore support the possibility that heterosynaptic plasticity exists at E-to-I synapses in CA3.

## Neuronal representations across hippocampal subregions during pattern separation and completion task

To validate our full model, we conducted 105 trials, varying the similarity between two patterns, A and B (Fig 11A and 11B). In each trial, pattern A was presented in a single 120 ms encoding phase, while input B was delivered over 30 theta cycles (7.2 s) that spanned both encoding and retrieval phases (Fig 11A).

To quantify discrimination, we averaged Δ output across all 30 retrieval phases. This revealed that each subregion transforms the inputs differently (Fig 11C). The DG exhibited robust pattern separation, with Δ output rising sharply once input dissimilarity (Δ input) exceeded ∼0.2, indicating that the DG cleanly separates inputs above a certain similarity threshold. This finding aligns with its ability to distinguish similar inputs [12,86]. By contrast, CA1 showed an almost linear Δ input-Δ output relationship, indicating a more gradual differentiation process. CA3 lies between these extremes. For Δ input < 0.2, its Δ output was lower than CA1's, indicating that CA3 seems to inherit the DG's inability to separate extremely

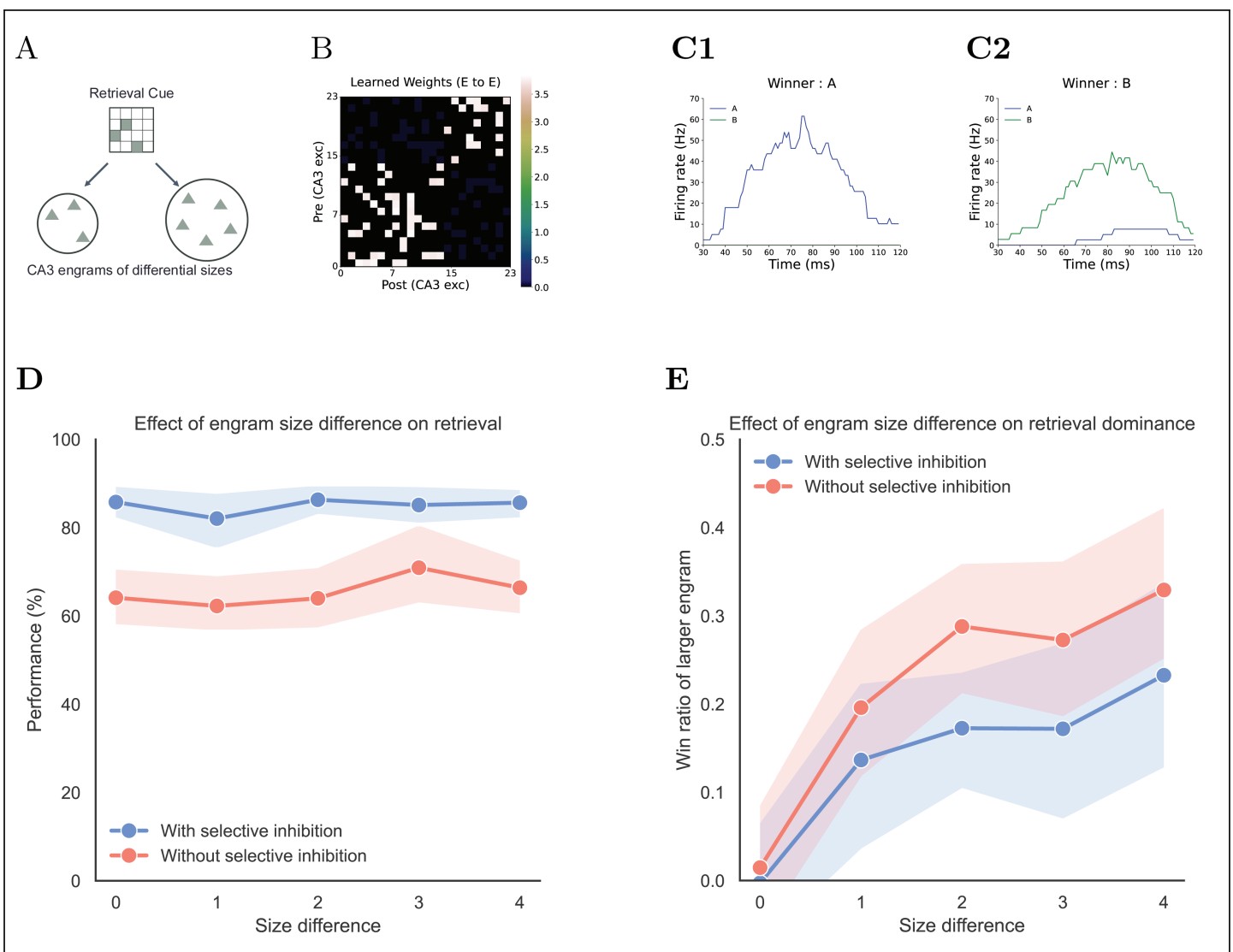

**Fig 10. Effect of engram size in CA3 on retrieval.** **(A)** Cue schematic: two CA3 engrams of different sizes are cued. **(B)** Example learned Rc weight matrix. **(C)** CA3 engram firing rates for example trials with the same cue: **(C1)** memory A is retrieved, **(C2)** memory B is retrieved. **(D)** Retrieval performance versus engram size differences, comparing conditions with (blue) and without (red) selective inhibition. **(E)** Dominance of the larger engram: fraction of successful trials retrieving the larger-memory engram, versus size differences, comparing with (blue) and without (red) selective inhibition.

similar inputs (Fig 11D). Yet for Δ input > 0.2, CA3's Δ output surpassed CA1's, suggesting that CA3 enhances separation once inputs are sufficiently distinct [10,14,16,87,88]. When we isolated retrieval phase responses, CA3's discrimination curve closely tracked CA1's (Fig 11E), in line with CA3 driving CA1 activity in our model. Notably, even at high input dissimilarity (Δ input ≈ 0.8), CA3's Δ output during retrieval remained below its encoding phase value. CA3 received a fully separated representation of pattern B from the DG during encoding, and pattern B was provided intact during retrieval. However, the competition from pattern A prevented CA3 from fully converging on the engram of pattern B. In summary, while the DG and CA3 achieve strong pattern separation during encoding, CA3's pattern completion process during retrieval is constrained by competition.

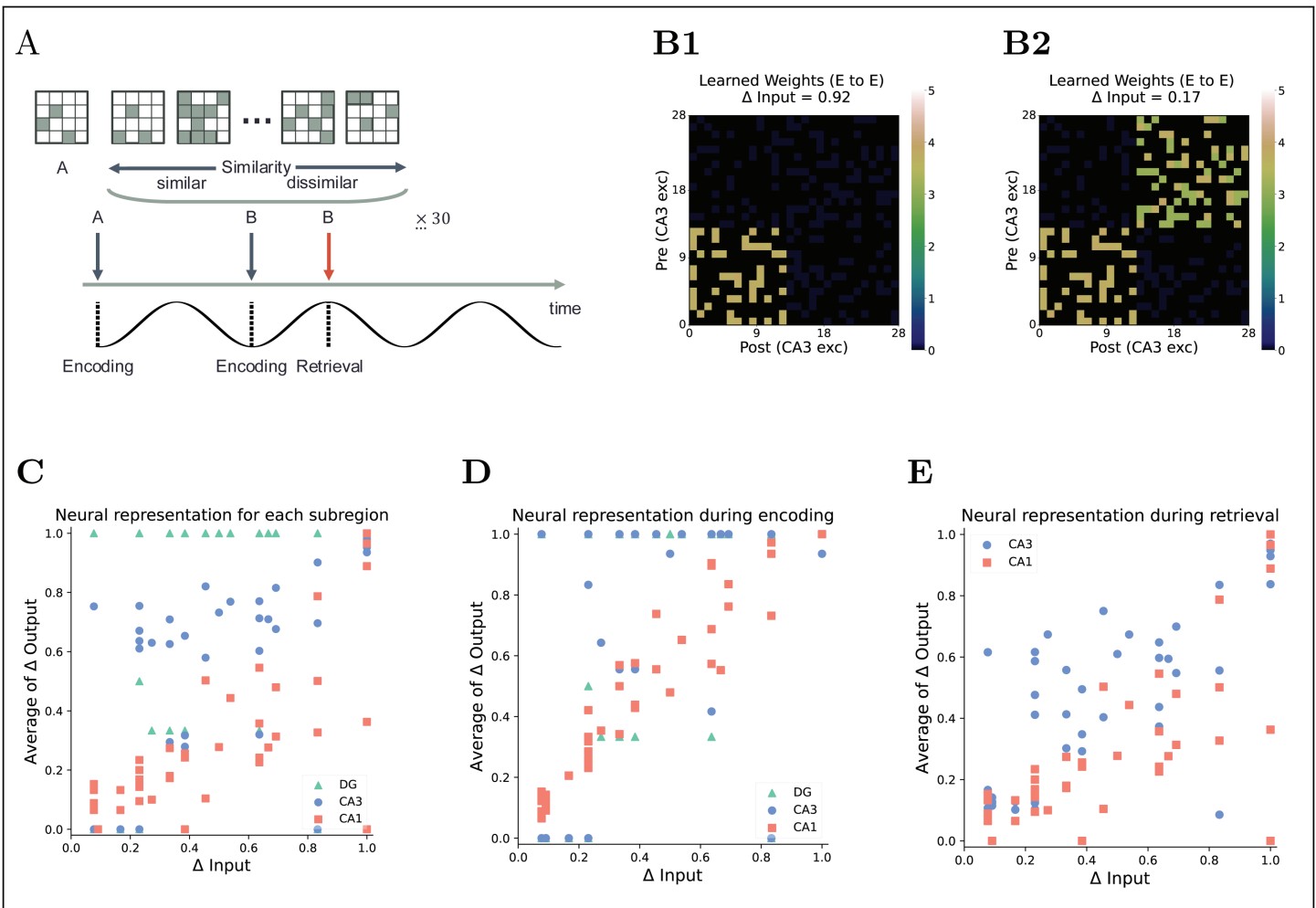

**Fig 11. Pattern separation task and results. (A)** Task schematic: Input A is presented in one 120-ms encoding phase. Input B is presented over 7.2 s (30 theta cycles spanning encoding and retrieval). B's similarity to A is varied across 105 trials. **(B)** Example learned Rc weight matrix for **(B1)** highly similar (Δ Input = 0.92) and **(B2)** less similar (Δ Input = 0.17) inputs. **(C)** Change in neural representation (Δ output) in each subregion versus input discrimination (Δ input), averaged over 30 retrieval phases. **(D)** Same Δ output vs. Δ input during encoding only. **(E)** Same during retrieval only.

To gain deeper insights into the dynamics of pattern completion, we created a graded cue task using two dissimilar stored memories, A and B (Fig 12B). During retrieval, we presented cues whose similarity to A and B varied linearly from 0 to 1 (Fig 12A). For example, if a cue was 70% similar to A, it would be 30% similar to B. This progression of cue bias allows us to map how retrieval accuracy shifts across the full continuum between two competing memories, rather than relying on arbitrary probe patterns. The results revealed a strong linear relationship between cue similarity and retrieval accuracy (Fig 12C and 12D). Because inputs A and B were completely dissimilar, each subregion produced nearly identical, proportional response curves. Notably, there was no abrupt shift to a single winning memory; even minimal cue bias yielded commensurate changes in recall. This proportional tuning shows that our model's pattern completion mechanism is highly sensitive and precise, allowing even weakly matching cues to drive accurate retrieval.

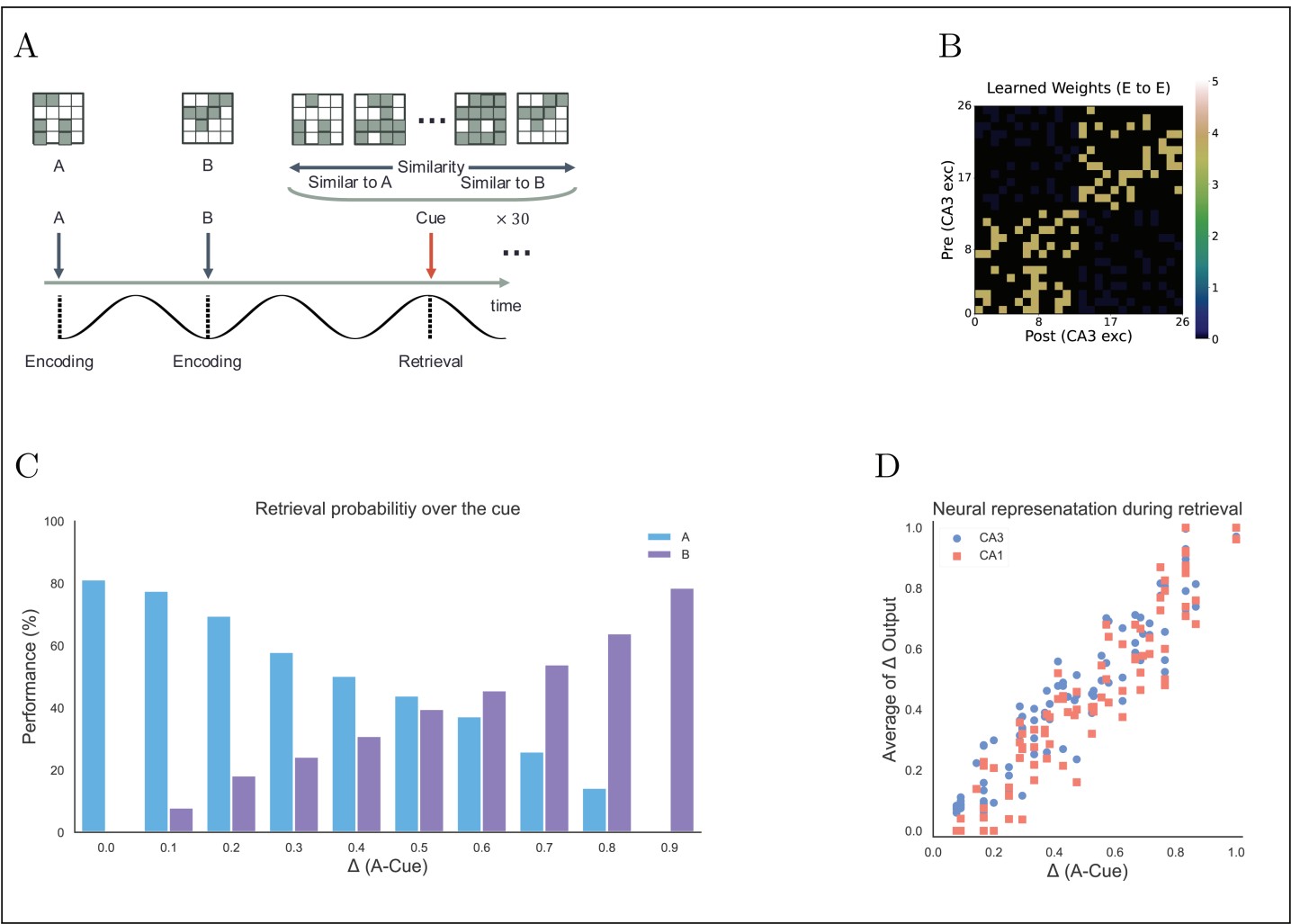

**Fig 12. Pattern completion task and results. (A)** Pattern completion task and results. Two orthogonal inputs (A and B, overlap = 0) are presented simultaneously during the encoding. During the retrieval, 100 cues are constructed with linearly varying similarity to A and B (overlaps sum to 1), each presented for 7.2 s (30 phases) without further learning. **(B)** Learned Rc weight matrix after training on A and B. **(C)** Retrieval performance versus cue similarity to A ($\Delta$ A–Cue). Performance is measured by the probability of retrieving A (blue) or B (purple) across 30 phases. **(D)** Change in neural representation ($\Delta$ output) versus cue discrimination (($\Delta$ A–Cue), averaged over 30 phases.

## Discussion

Using a spiking neural network model that captures the structural and connectivity properties of the hippocampus, we suggest an empirically veiled mechanism where CA3 inhibitory neurons are activated along with heterosynaptic plasticity at E-to-I synapses during encoding. This mechanism leads to selective inhibition during retrieval, enhancing the stability of pattern completion under the conditions where various shapes of engrams compete with each other. It occurs because only excitatory neurons, not suppressed by inhibitory neurons, survive to form neural assemblies. This mechanism for inhibitory neurons in memory processing is based on experimental evidence and theoretical assumptions, including (1) feedforward inhibition of the dentate gyrus via mossy fibers, (2) plasticity at inhibitory neurons, and (3) the segregation of encoding and retrieval phases by theta oscillation.

## Biological plausibility of the model

During encoding in the model, inhibitory neurons in CA3 receive inputs from the DG via mossy fibers. These inhibitory neurons then suppress most excitatory neurons in CA3, determining which neurons remain active (Fig 4). Some observations from the literature, although not all fully specific to the hippocampus, align with the proposed model; granule cells in the DG are known to recruit feedforward inhibition via mossy fibers [63,66,67]. Parvalbumin-positive (PV+) inhibitory neurons in the hippocampus are indeed crucial for forming and storing memory traces [22,31,32,89], and inhibitory neurons constrain the size of the engram [4,63–65,68]. These experimental findings underscore the importance of inhibitory neurons in memory formation, indicating that the model effectively captures their functional and structural roles.

Because our DG input is highly selective, each CA3 interneuron in the model is driven only by specific input patterns and thus inherits stimulus selectivity. This parallels findings in the cortex where certain interneurons exhibit tuned responses [23,26,27,90,91], implying hippocampal interneurons may carry feature-specific information. Growing evidence also shows that inhibitory synapses in the hippocampus undergo content-dependent plasticity during learning [28–34]. In practice, our model predicts that CA3 interneurons are not merely passive gain controllers but actively participate in both encoding and retrieval processes.

Our model assumes that the theta rhythm alternates the hippocampal circuit between two functional states—an encoding state centred on the trough of theta and a retrieval state centred on the peak [37,92–96]. Work in rodents and humans lends considerable support to this separation. In CA1, firing related to sensory encoding is strongest at the trough, whereas recall-related firing is strongest at the peak [97–99]. Similarly, [100] reported that CA3 population spiking during memory retrieval occurs preferentially at the CA1 theta peak. Mechanistic studies show that Rc is attenuated at the peak and direct PP input is attenuated at the trough, illustrating the principle that CA3 recurrent excitation and external drive are boosted at opposite phases [101]; most CA3 pyramidal cells are hyperpolarized during ongoing theta, reducing unspecific firing [102].

Although these findings align with the routing rule implemented in our model, direct causal demonstrations of phase-locked gating across the full trisynaptic loop are still sparse. In particular, these do not support a complete on/off switch for any hippocampal pathway. Synaptic efficacy in CA3 varies continuously with presynaptic burst frequency, local inhibition, and neuromodulatory state. For instance, low-frequency mossy fiber stimulation elicits primarily feed-forward inhibition, whereas theta-frequency bursts overcome that brake and robustly excite pyramidal cells [103]. The PP shows a comparable continuum of gain, sculpted by short-term plasticity and ongoing GABAergic feedback [104]. Consequently, the binary gating we impose on these two inputs should be viewed as a modeling abstraction that exaggerates the phase bias seen in vivo, enabling us to isolate the computational consequences of phase-segregated information flow.

## Alignment with cognitive evidence

Because selective inhibition enhanced model stability, we could examine firing patterns across hippocampal subregions during pattern separation and completion by varying input similarity. Distinct computational roles for hippocampal subregions are well documented [10,14]. The DG excels at pattern separation [8,11,12,86], while CA1 discrimination scales linearly with input differences [10,14,88]. CA3 exhibits more complex behavior, supporting both separation and completion, with experimental evidence favoring each role under different conditions [16,87,88].

The model reproduces these regional distinctions, showing strong separation in the DG and a strictly linear response in CA1. CA3 alternates between separation and completion in line with input similarity, matching empirical observations. Segregating encoding and retrieval phases allowed us to dissect CA3 dynamics: when inputs are highly similar, DG separation remains below threshold, so CA3 treats them as identical and performs completion, yielding more minor output differences than CA1.

Conversely, when inputs are dissimilar, CA3 separates them less than the DG but more than CA1. This attenuation reflects mixed outcomes. Cues can still trigger the completion of partially matching memories even when two are distinct. Retrieval probability scales linearly with cue–memory similarity, preventing dominance by any biased memory and capturing real-world complexity. Because the model aligns with cognitive data, it provides a valuable tool for studying hippocampal function, and its phase-specific analyses offer new insights into how the hippocampus resolves conflicting information.

## Experimental predictions

Identification of the model's key mechanism – heterosynaptic plasticity at E-to-I synapses, enabling targeted inhibition of competing engrams – allows us to propose explicit, testable predictions. While certain in silico manipulations are not feasible in vivo, modern techniques can achieve many analogous experiments. For instance, neurons active during memory formation can be selectively labeled via TRAP or Fos-tet tagging [105,106], and specific cell populations or synapses can be perturbed using optogenetic or chemogenetic tools [107]. Likewise, calcium imaging and multielectrode recordings now permit readout of engram reactivation with single-cell resolution during behavior [108,109]. We use these approaches to outline three major predictions that translate our model's mechanism into concrete experiments.

First, to verify memory-specific plasticity onto interneurons, one can combine three complementary approaches. Fluorescence co-labeling of recently active CA3 cells will reveal whether a subset of interneurons is repeatedly coactivated with the excitatory engram. In vivo two-photon calcium imaging can track these interneurons across learning and retrieval, asking whether their activity increases with the tagged excitatory ensemble. Finally, targeted whole-cell recordings from tagged interneurons in acute slices, while stimulating dentate gyrus mossy fibers, can measure any learning-induced potentiation of excitatory postsynaptic currents relative to naïve controls. If heterosynaptic potentiation has occurred, excitatory postsynaptic currents onto these interneurons will be significantly larger than in naive controls or non-tagged interneurons. Demonstrating such potentiation would confirm the synaptic imprint predicted by the model.

Second, the model suggests that a subset of inhibitory neurons recruited during encoding is necessary to improve the competition dynamics that ensure the appropriate memory engram dominates during retrieval. An intersectional tagging strategy can isolate them, combining activity-dependent labelling with a GABAergic driver line (TRAP plus PV-Cre or GAD2-Cre) so that only inhibitory engram cells express an optogenetic or chemogenetic effector. If the tagged inhibitory is critical, such an intervention should cause the animal to confuse similar memories or retrieve spurious associations, as the loser engram is no longer actively inhibited. Notably, this prediction mirrors the logic of classical engram experiments on excitatory neurons; just as silencing excitatory engram cells impairs memory recall, we propose that removing the corresponding inhibitory engram cells will impair the ability to resolve competition between memories. This can be verified by monitoring network activity with high-density electrophysiology or two-photon imaging during recall trials. Such

results would confirm that interneurons active at encoding are indispensable for the selective retrieval of that memory.

A further prediction is that the level of inhibition present during memory encoding will determine the size of the excitatory engram, which in turn can bias retrieval dominance, and that our proposed selective inhibitory mechanism counteracts this bias. This can be tested by experimentally manipulating interneuron activity during memory formation to create small vs large engrams and then examining recall outcomes with and without selective inhibition. If inhibitory tone is artificially reduced during learning, more principal cells will likely be recruited into the engram than usual. Conversely, artificially increasing inhibition during encoding would constrain the engram to fewer neurons.

One could use a chemogenetic approach to dampen global CA3 inhibition during the learning of context A to yield an unusually large engram for that context while leaving inhibition intact for the learning of Context B. Later, when presenting a partial cue for context B, one would measure which engram is reactivated in animals with intact circuitry vs animals in which heterosynaptic E-to-I plasticity had been blocked. Our model expects that in control animals, the appropriate engram B is recalled despite its smaller size, because its specific inhibitory connections suppress the larger engram A. Still, if the inhibitory plasticity is blocked, the abnormally large engram A will tend to intrude on or dominate the recall of B. This qualitative outcome would demonstrate that inhibitory plasticity normally mitigates size-based recall biases. Notably, findings in other memory systems lend support to these idea: In the dentate gyrus, lateral inhibition via somatostatin interneurons regulates the number of granule cells recruited into an engram and stabilizes memory recall [65], and [64] transiently silenced PV+ inhibitory neurons in the lateral amygdala during learning and thereby enlarged the ensuing memory engram. Our prediction generalizes this principle to CA3, positing that by tailoring inhibition to each engram, the brain ensures that bigger memories do not always out-compete smaller ones, provided the cue uniquely matches the smaller engram.

## Previous models of hippocampus

Several prior computational models of the hippocampus have combined full-circuit simulations with theta-rhythm phase dynamics to improve memory encoding and retrieval beyond simple Hebbian learning. For example, [110] introduced a model in which alternating theta phases (encoding vs. recall) produce a powerful error-driven learning rule, significantly expanding storage capacity relative to classical Hebbian models. In that model, the difference between a target input pattern (imposed during one theta phase) and the network's attempted recall of that pattern (from a previous phase) serves as an error signal that drives synaptic adjustments.

[111] likewise leveraged theta-phase segregation in a full hippocampal network. Their model instantiated distinct theta states corresponding to encoding (dominated by entorhinal-to-CA1 input) versus retrieval (dominated by CA3-to-CA1 input). By combining Hebbian and error-driven learning across these phases, the system concurrently supported the memorization of specific episodes and the extraction of cross-episode regularities. In particular, that model's direct monosynaptic entorhinal-CA1 pathway captured statistical regularities, while the trisynaptic pathway encoded individual episodic memories, exemplifying a complementary learning system within the hippocampus.

More recent work extends the error-driven framework deeper into CA3. [112] developed the Theremin model, which introduced an error-driven learning mechanism in the CA3. In their model, CA3 receives a brief entorhinal input before a stronger DG input arrives, creating a temporal difference between the initial and subsequent activation states. This

difference serves as a teaching signal. The later DG target pattern drives synaptic changes in CA3 (including CA3 recurrent and EC to CA3 synapses), yielding more pattern-separated memory representations. [112] showed that using DG as a teacher markedly reduces interference and accelerates learning relative to a purely Hebbian baseline. Crucially, they argued that this error signal could be implemented via heterosynaptic plasticity—DG activation gating adjustments at other EC to CA3 synapses—giving their rule a plausible biological substrate.

A common thread in these error-driven models is cross-phase interaction. Weights are updated by comparing activity patterns across successive theta half-cycles. Our model takes a distinct approach by fully segregating encoding and retrieval phases. While adhering to the theta framework, this deliberate separation simplifies the analysis of retrieval-phase dynamics, letting us focus on how engram-specific inhibitory neurons suppress competing traces during recall. Such an inhibitory gating strategy provides new insights into the contributions of inhibitory circuits to memory stability, complementing the weight-based error correction mechanisms emphasized in other error-driven models.

## Conclusion

Our research highlights the inhibitory neurons' active and dynamic role in CA3 engram formation and stable pattern completion. This work provides a fresh perspective on established hippocampal functions and opens new avenues for experimental investigation. As our understanding of hippocampal circuitry and function continues to evolve, models like ours will be essential in bridging the gap between cellular mechanisms and cognitive phenomena, advancing neuroscience and artificial intelligence research.

## Materials and methods

### Model structure

We developed a spiking neural network model of the entorhinal–hippocampal system, including superficial and deep layers of the EC and the hippocampal subregions DG, CA3, and CA1. The network emphasizes the trisynaptic pathway while simplifying the direct EC–CA1 projection. The DG region was modeled with two layers (hilus and the GCL), each containing excitatory and inhibitory neurons, analogous to the cellular composition of CA3. Memory formation and competition occur primarily in CA3 in this architecture, with key projections retained. For example, the mossy fiber inputs from DG to CA3 and the direct PP inputs from superficial EC to CA3 are included.

Given the focus on spatial patterns, we considered only spatial pattern separation in DG, excluding temporal and rate-based separation mechanisms. Because we ignore temporal coding, this assumption leads to some departures from biological realism: for example, neuron population sizes, connection densities, and synaptic weights are adjusted from experimental values (Tables 1 and 2). Initial parameters were derived from the study by [113] and subsequently optimized through iterative manual adjustments to ensure proper model functioning. Connection weights were initialized with specific values and fixed to preserve connection probabilities during learning. Instead of adjusting these weights directly, we modulated the peak conductance, which influences synaptic efficacy (detailed explanation in the next section).

The input layer (superficial EC) consists of 16 neurons, yielding $2^{16}$ unique spatial input patterns. This configuration provides sufficient pattern variability for testing the model while keeping simulations tractable. Each superficial EC neuron was driven with a sustained 50 Hz

**Table 1. Parameters for neurons organizing each layer.**

| | N | $\tau_{AMPA}$ | $\tau_{NMDA}$ | $\tau_{GABA_A}$ | $\tau_{GABA_B}$ | $W_{Be}$ | $W_{Bi}$ |
|---|---|---|---|---|---|---|---|
| Superficial EC | 16 | 5 | - | - | - | 5 | - |
| Deep EC | 16 | 5 | - | - | - | 5 | - |
| Hilus (exc) | 100 | 5 | - | 15 | - | 5 | 10 |
| Hilus (inh) | 16 | 5 | - | - | - | 5 | - |
| GCL (exc) | 800 | 5 | - | 15 | - | 5 | 10 |
| GCL (inh) | 400 | 5 | - | - | - | 7 | 7 |
| CA3 (exc) | 2400 | 5 | 30 | 8 | 30 | 4, 0.75, 20 | 10 |
| CA3 (inh) | 120 | 5 | 30 | 8 | 30 | 5 | 10 |
| CA1 | 200 | 5 | 30 | - | - | 4 | - |

**Table 2. Parameters for connections between layers.**

| | $p_{conn}$ | W | $t_d$ | Activation Phase | Learning |
|---|---|---|---|---|---|
| Superficial EC→Hilus (exc) | 0.3125 | 3 | - | E | - |
| Superficial EC→Hilus (inh) | One-to-One | 3 | - | E | - |
| Superficial EC→GCL (exc) | 0.0625 | 2 | 10 | E | - |
| Superficial EC→GCL (inh) | 0.5 | 4 | - | E | - |
| Superficial EC→CA3 (exc) | 0.25 | 1 | - | R | O |
| Superficial EC→Deep EC | One-to-One | 3 | - | E | - |
| Deep EC→CA1 | 0.125 | 0.5 | - | E | - |
| Hilus (inh)→Hilus (exc) | 0.0625 | 4 | - | E | - |
| Hilus (exc)→GCL (inh) | 0.5 | 4 | - | E | - |
| GCL (inh)→GCL (exc) | 0.0013 | 3 | - | E | - |
| GCL (exc)→CA3 (exc) | 0.0125 | 2 | 3 | E | - |
| GCL (exc)→CA3 (inh) | 0.025 | 2 | - | E | - |
| CA3 (exc)→CA3 (exc) | 0.25 | 1 | 5 | R | O |
| CA3 (exc)→CA3 (inh) | 0.25 | 1 | - | R | O |
| CA3 (exc)→CA1 | 0.5 | 0.15 | - | R | O |
| CA3 (inh)→CA3 (exc) | 0.25 | 2 | - | A | - |
| CA3 (inh)→CA3 (inh) | 0.167 | 0.5 | - | A | - |
| CA1→Deep EC | 0.125 | 2 | 15 | R | - |
| Noise→CA3 (exc) | - | 0.5 | - | R | - |

Activation Phase indicates the connection status: E represents activation during the encoding phase, R during the retrieval phase, and A for activation across all phases. Physical dimension for $t_d$ is ms.

spike train. We clarify that this 50 Hz drive is a modeling simplification rather than a physiological claim. In reality, EC neurons exhibit gamma-band oscillations ($\sim$30–100 Hz), and individual cells fire only briefly at each cycle [114]. Thus, the 50 Hz input should be seen as an approximation of the strong excitatory effect of entorhinal gamma rhythms, ensuring that CA3 receives enough excitatory drive to form stable engrams.

The CA1 region in the model is simplified to include only excitatory neurons (Table 1). Connections between CA1 and deep EC were designed so that each CA1 neuron maintains consistent bidirectional connections with specific deep EC neurons, based on a defined connection probability (Table 2). Additionally, deep EC receives inputs from superficial EC via a one-to-one connection, providing input information to CA1 during encoding. While this simplification omits some intricate microcircuitry in biological CA1 and its connections with EC [115,116], it retains the essential function of interfacing between CA3 and deep EC, allowing CA3 engrams to effectively encode and decode information between hippocampal and cortical representations.

We implemented the model in Python using NumPy. The following sections describe the specific neuron models, learning rules, and analysis methods used to investigate memory dynamics in this network.

### Single neuron models

Each neuron was described using the Izhikevich neuron model [117] as:

$$C\dot{v} = k(v - v_r)(v - v_t) - u - I_{syn} \tag{1}$$

$$\dot{u} = a\{b(v - v_r) - u\} \text{ if } v \geq 30mV, \text{ then } v = c, u = u + d \tag{2}$$

where $C$ is the membrane capacitance, $v$ is the membrane potential, $v_r$ is the resting potential, $v_t$ is the threshold potential, and $u$ is the recovery variable (the difference of all inward and outward voltage-gated currents). $I_{syn}$ is the total synaptic input current, detailed in the next section. We used distinct Izhikevich parameter sets containing $a$, $b$, $c$, $d$, and $k$ for each region's excitatory and inhibitory neuron types, based on published values (Table 3). These parameters were chosen to produce firing characteristics appropriate for each neuron class: regular-spiking excitatory cells and fast-spiking inhibitory interneurons.

### Synapse models

Synaptic currents were modeled with fast and slow receptor dynamics for each connection, following [113,117]. Each synapse included up to four conductance-based components: AMPA and NMDA (excitatory neurotransmission) and GABA$_A$, and GABA$_B$ (inhibitory transmission). The total synaptic current from a postsynaptic neuron's perspective was calculated as:

$$I_{syn} = g_{AMPA}(v - 0) + g_{NMDA}\frac{[(v + 80)/60]^2}{1 + [(v + 80)/60]^2}(v - 0) + g_{GABA_A}(v + 70) + g_{GABA_B}(v + 90) \tag{3}$$

$$\begin{cases} \dfrac{dg_i^{exc}}{dt} = -\dfrac{g_i^{exc}}{\tau_{exc}} + \sum_{j \in exc} q_{ij}w_{ij}S_j(t), \quad S_j(t) = \sum_k \delta(t - t_j^t). \\ \dfrac{dg_i^{inh}}{dt} = -\dfrac{g_i^{inh}}{\tau_{inh}} + \sum_{j \in inh} q_{ij}w_{ij}S_j(t). \end{cases} \tag{4}$$

where excitatory reversal potentials are set to 0 mV, and inhibitory reversal potentials are set to –70 mV and –90 mV, respectively. The ratio of NMDA to AMPA receptors was set to 5:5, and the ratio of GABA$_A$ to GABA$_B$ receptors was set to 9:1 for all of the neurons in CA3 [118]. A simplified voltage-dependent gating factor $[(v + 80)/60]^2/[1 + ((v + 80)/60)^2]$ was applied to NMDA currents to account for Mg$^{2+}$ block. The AMPA and GABA$_A$ synaptic

**Table 3. Parameters of Izhikevich model for excitatory neurons and inhibitory neurons.**

|     | $C$ | $k$ | $V_{rest}$ | $V_{th}$ | $V_{peak}$ | $a$ | $d$ | $c$ | $d$ |
|-----|-----|-----|-----------|----------|-----------|-----|-----|-----|-----|
| Exc | 80  | 3   | -60       | -50      | 50        | 0.01 | 5   | -60 | 10  |
| Inh | 20  | 1   | -55       | -40      | 25        | 0.15 | 8   | -55 | 200 |

conductances were modeled as fast-decaying components, whereas NMDA and GABA$_B$ were slower. And for simplification, the synapse models of neurons in the DG included only AMPA and GABA$_A$ receptors because learning does not occur in this region of the model (Table 1).

Synaptic conductances onto neuron i evolve as the sum of two processes: an exponential decay and instantaneous increases driven by presynaptic spikes. Between spikes, each conductance $g_i$ decays toward zero with its own time constant, $\tau$. Whenever a presynaptic neuron j fires at time $t_j^k$, the corresponding conductance $g_i$ is incremented by the product of the synaptic weight $w_{ij}$ and the peak conductance $q_{ij}$, with the spike represented as a Dirac delta function [119]. For connections without learning, the peak conductance for non-plastic connections $q$ is initially set to 3 nS. For connections with learning, peak conductance $q$ is initially set to 0 nS except for E-to-I connections in CA3, where $q$ is set to 0.5 nS to establish global inhibition as the default state.

The total excitatory and inhibitory synaptic currents were limited to maximum values of $W_{Be}$ and $W_{Bi}$, respectively (Table 1). For excitatory neurons in CA3, the total synaptic current was differentiated based on the region of origin of the sending neuron—superficial EC, DG, or CA3—to preserve the distinct influences of each region. Additionally, some connections featured delayed transmission times to the postsynaptic neuron to ensure stable learning (Table 2).

## STDP learning and heterosynaptic plasticity

We implemented learning using the symmetric STDP rule. This learning rule was based on experimental observations from CA3 neuron pairs [55,57]. In line with the conductance-based STDP model of [119], we implemented plasticity by adjusting the peak synaptic conductance $q$ rather than the abstract weight parameter $w$. The synaptic weights were initialized and fixed to preserve the network's original connectivity and stability. (Table 2). Under this scheme, STDP modifies only the strength of existing synapses, ensuring that the overall connection probability is unchanged. Because our STDP rule implements potentiation only, the network cannot self-normalize its weight distribution like a balanced (LTP + LTD) rule would [120]. We also silenced recurrent activity during encoding, eliminating the spike-timing competition that might otherwise promote homeostatic weight normalization. Under these combined assumptions, the post-learning distribution of synaptic strengths is not expected to mirror the pre-learning distribution.

Synaptic weights evolve as follows:

$$q = q + F(\Delta t), \ F(\Delta t) = \begin{cases} A_+ \exp\left(-\dfrac{\Delta t}{\tau_+}\right) & \text{at } t_{post}, \text{ if } t_{pre} < t_{post} \\ A_- \exp\left(\dfrac{\Delta t}{\tau_-}\right) & \text{at } t_{pre}, \text{ if } t_{post} < t_{pre} \end{cases} \tag{5}$$

where $\Delta t = t_{post} - t_{pre}$ represents the time difference between action potentials, and $A_\pm$ determines the maximum amount of synaptic modification, which decays exponentially with time constants $\tau_\pm$. Given the symmetric nature of the adopted STDP rule, $A_\pm$ was set to 0.2 nS, and $\tau_\pm$ was set to 62.5 ms. Similar to general synaptic plasticity, when the time window between presynaptic and postsynaptic spikes is sufficiently short, the peak conductance is incremented by $F(\Delta t)$. However, with the adoption of heterosynaptic plasticity, the presynaptic stimulus no longer directly influences postsynaptic firing. The peak conductance is constrained within the range $[0, q_{max}]$, where $q_{max} = 3$ nS. The connections where learning occurs are listed in Table 2.

## Similarity analysis

To quantify the network's ability to separate or complete activity patterns, we measured the similarity between output firing patterns using an overlap metric [79]. For any two patterns $i$ and $j$, the overlap $O_{ij}$ was defined as:

$$O_{ij} = \frac{n_{ij}}{n_i + n_j} \tag{6}$$

where $n_i$ represents the number of activated neurons in pattern $i$, and $n_{ij}$ denotes the number of neurons activated in both patterns $i$ and $j$. A value $O_{ij} = 1$ indicates identical firing ensembles, whereas $O_{ij} = 0$ means the two patterns share no active neurons. We further defined a discrimination index $\Delta_{ij} = 1 - O_{ij}$, so that more distinct patterns yield higher $\Delta$ (up to 1.0) while overlapping patterns have lower values.

When evaluating pattern retrieval in the CA3 network, we applied a firing-rate threshold to distinguish meaningful activity from noise. In CA3 output patterns, only neurons firing above 25 Hz during retrieval were considered active. Neurons firing at lower rates were treated as inactive to filter out spurious or background firing. This thresholding ensured pattern overlap calculations reflected only robust, sustained activity, not transient or random spiking. Although this sustained 25 Hz activity exceeds the ∼0.3–-5 Hz typically reported for CA3 pyramidal neurons in vivo, we use it solely as an analysis criterion to sharpen the distinction between winning and losing engrams.

## Author contributions

**Conceptualization:** Gyeongtae Kim.

**Data curation:** Gyeongtae Kim.

**Formal analysis:** Gyeongtae Kim.

**Funding acquisition:** Pilwon Kim.

**Investigation:** Gyeongtae Kim.

**Methodology:** Gyeongtae Kim.

**Software:** Gyeongtae Kim.

**Supervision:** Pilwon Kim.

**Validation:** Gyeongtae Kim.

**Visualization:** Gyeongtae Kim.

**Writing – original draft:** Gyeongtae Kim.

**Writing – review & editing:** Gyeongtae Kim, Pilwon Kim.

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
