## [Decision Letter · Decision Letter 0]

27 Mar 2025

PCOMPBIOL-D-24-01892

Selective inhibition in CA3: A mechanism for stable pattern completion through heterosynaptic plasticity

PLOS Computational Biology

Dear Dr. Kim,

Thank you for submitting your manuscript to PLOS Computational Biology. After careful consideration, we feel that it has merit but does not fully meet PLOS Computational Biology's publication criteria as it currently stands. Therefore, we invite you to submit a revised version of the manuscript that addresses the points raised during the review process.

Please submit your revised manuscript within 30 days May 27 2025 11:59PM. If you will need more time than this to complete your revisions, please reply to this message or contact the journal office at ploscompbiol@plos.org. Please include the following items when submitting your revised manuscript:

We look forward to receiving your revised manuscript.

Kind regards,

Daniel Bush

Academic Editor

PLOS Computational Biology

Hugues Berry

Section Editor

PLOS Computational Biology

**Additional Editor Comments :**

The reviewers have each requested that some additional technical details of the simulations and results be included. In addition (as also noted by Reviewer 2), it is crucial that this model makes testable predictions for future experimental work, as well as setting these results in the appropriate context of previous work, and being upfront about any limitations to their biological plausibility.

**Journal Requirements:**

2) Thank you for stating "The dataset supporting these findings is available on Zenodo at https://zenodo.org/uploads/14016721." We couldn't access the dataset through the provided link. Please provide us with a new link or provide further details to locate the data."

3) Please amend your detailed Financial Disclosure statement. This is published with the article. It must therefore be completed in full sentences and contain the exact wording you wish to be published. 

3) If any authors received a salary from any of your funders, please state which authors and which funders.

4) The file inventory includes multiple files for Figures 1,2, and (4-12). We would recommend either combining these into single Figure .tiff files with separate internal panels, or renumbering them as individual figures, as we are not able to publish multiple components of a single figure as separate files. Please also ensure that the figures are uploaded in a correct numerical order in the online submission form.

**Reviewers' comments:**

Reviewer's Responses to Questions

**Comments to the Authors:**

**Please note that one of the reviews is uploaded as an attachment.**

Reviewer #1: Uploaded as an attachment

Reviewer #2: This paper presents a compelling computational model investigating the role of selective inhibition in CA3 and its effects on pattern completion, in contrast to traditional models emphasizing global inhibition. The modeling approach is rigorous, and the results convincingly demonstrate the benefits of selective inhibition in stabilizing memory retrieval. The study is well-structured, and the extensive analyses lend credibility to the proposed mechanism.

However, I have several concerns regarding the biological plausibility of some of the model’s assumptions, particularly regarding the theta-phase encoding/retrieval framework and the on/off switching of CA3 projections. Additionally, the paper does not sufficiently situate its findings within the broader literature, especially in relation to similar models of hippocampal memory. Below, I outline my major concerns and suggestions for improvement.

Major Comments

1. Lack of Empirical Justification for Theta-Modulated Trisynaptic Pathway

• The authors assume that the trisynaptic pathway (EC → DG → CA3) follows theta phase-dependent switching between encoding and retrieval, but no direct empirical evidence is provided for this assumption.

• While theta oscillations have been extensively studied in CA1, their role in gating CA3 input-output dynamics is less well established. The authors should either cite relevant experimental studies or clarify that this remains an open question.

2. On/Off Switching of CA3 Projections Lacks Justification

• A key component of the model is the phase-dependent activation and silencing of CA3 pathways (e.g., recurrent collaterals (Rc), perforant path (PP), and Schaffer collaterals (Sc)).

• The authors do not provide any direct experimental evidence for this mechanistic switching. How does this align with known hippocampal circuit physiology?

• If this is a theoretical assumption rather than an experimentally supported fact, the authors should clearly state this and discuss its potential limitations.

3. Lack of Testable Predictions & Experimental Implications

• The paper presents strong modeling results, but no testable predictions are provided.

• Given the novelty of the selective inhibition framework, the authors should outline potential experimental tests that could validate their model, such as:

• Optogenetic inhibition of CA3 interneurons during retrieval to test whether competition between engrams is disrupted.

• In vivo electrophysiology to assess whether CA3 inhibitory activity exhibits phase-dependent selectivity.

4. Connections to Prior Work Are Missing

• The proposed model closely parallels existing models that incorporate theta-modulated hippocampal dynamics and heterosynaptic plasticity. A particularly relevant study is:

• Zheng et al. (2022): “Correcting the Hebbian Mistake: Toward a Fully Error-Driven Hippocampus” (PLoS Comput. Biol.).

• This prior work also explores how CA3 representations are constrained by DG activity using a theta-phase framework, making it highly relevant to the present study.

• The presented paper has substantial significance in its biological plausibility of selectivity in inhibitory neurons. The authors should explicitly compare their findings to these existing models and discuss where their work provides new insights or improvements.

Minor Comments & Suggestions

• The paper contains redundant text in several sections, making the manuscript longer than necessary.

• The authors should carefully edit for conciseness and coherence, particularly in the Results and Discussion sections.

• Sentence structure and punctuation should be reviewed for clarity and readability.

Final Recommendation

The study presents a well-constructed model with significant theoretical implications for hippocampal memory dynamics. However, before publication, the authors should:

1. Provide stronger empirical justification for the theta-gated trisynaptic model.

2. Clarify the biological basis for the phase-dependent on/off switching of CA3 pathways.

3. Include testable predictions and discuss how their model could be experimentally validated.

4. Compare their findings to existing hippocampal models, particularly Zheng et al. (2022).

5. Revise the manuscript for clarity by eliminating redundancy and improving text coherence.

Addressing these concerns will significantly strengthen the impact of the paper and its relevance to the broader hippocampal modeling community.

**Have the authors made all data and (if applicable) computational code underlying the findings in their manuscript fully available?**

Reviewer #1: Yes

Reviewer #2: Yes

PLOS authors have the option to publish the peer review history of their article (what does this mean?). If published, this will include your full peer review and any attached files.

Reviewer #1: No

Reviewer #2: No

**Figure resubmission:**
---

## [Decision Letter · Decision Letter 1]

24 Jun 2025

Dear Professor Kim,

We are pleased to inform you that your manuscript 'Selective inhibition in CA3: A mechanism for stable pattern completion through heterosynaptic plasticity' has been provisionally accepted for publication in PLOS Computational Biology.

Best regards,

Daniel Bush

Academic Editor

PLOS Computational Biology

Hugues Berry

Section Editor

PLOS Computational Biology

Reviewer's Responses to Questions

**Comments to the Authors:**

Reviewer #1: Thank you for addressing the comments raised by me. It would be nice to add references to some of the methods suggested in Experimental predictions sections so that the readers can be directed to it easily.

Reviewer #2: The authors' detailed responses have addressed all my previous questions/comments and I'd recommend accepting the paper as it is.

**Have the authors made all data and (if applicable) computational code underlying the findings in their manuscript fully available?**

Reviewer #1: Yes

Reviewer #2: Yes

PLOS authors have the option to publish the peer review history of their article (what does this mean?). If published, this will include your full peer review and any attached files.

Reviewer #1: No

Reviewer #2: No

---

## [Editor Report · Acceptance letter]

29 May 2025

PCOMPBIOL-D-24-01892R1

Selective inhibition in CA3: A mechanism for stable pattern completion through heterosynaptic plasticity

Dear Dr Kim,

I am pleased to inform you that your manuscript has been formally accepted for publication in PLOS Computational Biology. Your manuscript is now with our production department and you will be notified of the publication date in due course.

With kind regards,

Zsofia Freund
